# Structural insights into the molecular effects of the anthelmintics monepantel and betaine on the *Caenorhabditis elegans* acetylcholine receptor ACR-23

Fenglian Liu [1,8], Tianyu Li [2,3,8], Huihui Gong[1,8], Fei Tian[1], Yan Bai[1], Haowei Wang [1], Chonglin Yang [1], Yang Li [3,4,5], Fei Guo [3,4,5✉], Sheng Liu [6,7✉] & Qingfeng Chen [1✉]

## Abstract

**Anthelmintics are drugs used for controlling pathogenic helminths in animals and plants. The natural compound betaine and the recently developed synthetic compound monepantel are both anthelmintics that target the acetylcholine receptor ACR-23 and its homologs in nematodes. Here, we present cryo-electron microscopy structures of ACR-23 in apo, betaine-bound, and betaine- and monepantel-bound states. We show that ACR-23 forms a homo-pentameric channel, similar to some other pentameric ligand-gated ion channels (pLGICs). While betaine molecules are bound to the classical neurotransmitter sites in the inter-subunit interfaces in the extracellular domain, monepantel molecules are bound to allosteric sites formed in the inter-subunit interfaces in the transmembrane domain of the receptor. Although the pore remains closed in betaine-bound state, monepantel binding results in an open channel by wedging into the cleft between the transmembrane domains of two neighboring subunits, which causes dilation of the ion conduction pore. By combining structural analyses with site-directed mutagenesis, electrophysiology and in vivo locomotion assays, we provide insights into the mechanism of action of the anthelmintics monepantel and betaine.**

**Keywords** Acetylcholine Receptor; Monepantel; Betaine; Anthelmintics; Cryo-EM Structure
**Subject Categories** Microbiology, Virology & Host Pathogen Interaction; Pharmacology & Drug Discovery; Structural Biology

## Introduction

As well as being a direct threat to human health, pathogenic helminths also cause economic losses by compromising crop and livestock production. As few vaccines are available, anthelmintics have been primarily used for control of pathogenic helminths, with a global market size of 2.6 billion USD in 2023 (https://www.transparencymarketresearch.com/anthelmintic-drugs-market.html). In the past decades, control of pathogenic helminths has relied heavily on three major classes of anthelmintics —the benzimidazoles, imidazothiazoles and macrocyclic lactones, which inevitably resulted in drug resistance to these anthelmintics (Geerts and Gryseels, 2000). The amino-acetonitrile derivatives (AADs) are a new chemical class of synthetic anthelmintics developed recently, which showed a different mode of action as compared to other anthelmintics, and great efficacy towards parasitic nematodes that are resistant to traditional anthelmintics (Ducray et al, 2008; Kaminsky et al, 2008a; Kaminsky et al, 2008b; Prichard and Geary, 2008). One of the AADs, monepantel, has been approved for controlling gastrointestinal worms in sheep, under the trademark of Zolvix. However, drug resistance for monepantel has begun to appear (Scott et al, 2013). Therefore, a better understanding of its mechanism of action will facilitate future efforts in drug development to overcome the emerging problems of drug resistance.

Similar to most other synthetic anthelmintics such as midazothiazoles and macrocyclic lactones, which target acetylcholine-gated ion channels and glutamate-gated chloride channels, respectively (Raisch and Raunser, 2023), monepantel also mainly targets Cys-loop family ligand-gated ion channels (LGICs), e.g., acetylcholine receptor like-23 (ACR-23) from the model nematode *Caenorhabditis elegans*, which is a member of nematode specific DEG-3 subfamily of Cys-loop family pLGICs (Kaminsky et al, 2008a; Rufener et al, 2009). As an evidence, 27 out of 44 AADs resistant alleles identified in *Caenorhabditis elegans* were on ACR-23 (Kaminsky et al, 2008a). In addition, disruption of an ACR-23 homolog in the parasitic nematode *Haemonchus contortus*,

[1]Center for Life Sciences, Yunnan Key Laboratory of Cell Metabolism and Diseases, State Key Laboratory for Conservation and Utilization of Bio-Resources in Yunnan, School of Life Sciences, Yunnan University, Kunming 650091, China. [2]Shanghai Key Laboratory of Maternal Fetal Medicine, Shanghai Institute of Maternal-Fetal Medicine and Gynecologic Oncology, Department of Anesthesiology, Shanghai First Maternity and Infant Hospital, School of Medicine, Tongji University, Shanghai 201204, China. [3]State Key Laboratory of Drug Research, Shanghai Institute of Materia Medica, Chinese Academy of Sciences, Shanghai 201203, China. [4]University of Chinese Academy of Sciences, Beijing 100049, China. [5]National Clinical Research Center for Aging and Medicine, Huashan Hospital, Fudan University, Shanghai 200040, China. [6]Institute of Pediatrics, Shenzhen Children's Hospital, Shenzhen, Guangdong Province 518026, China. [7]Department of Infectious Diseases, Shenzhen Children's Hospital, Shenzhen, Guangdong Province 518038, China. [8]These authors contributed equally: Fenglian Liu, Tianyu Li, Huihui Gong. ✉E-mail: guofei@simm.ac.cn; sliu520@mail.ustc.edu.cn; qingfengchen@ynu.edu.cn

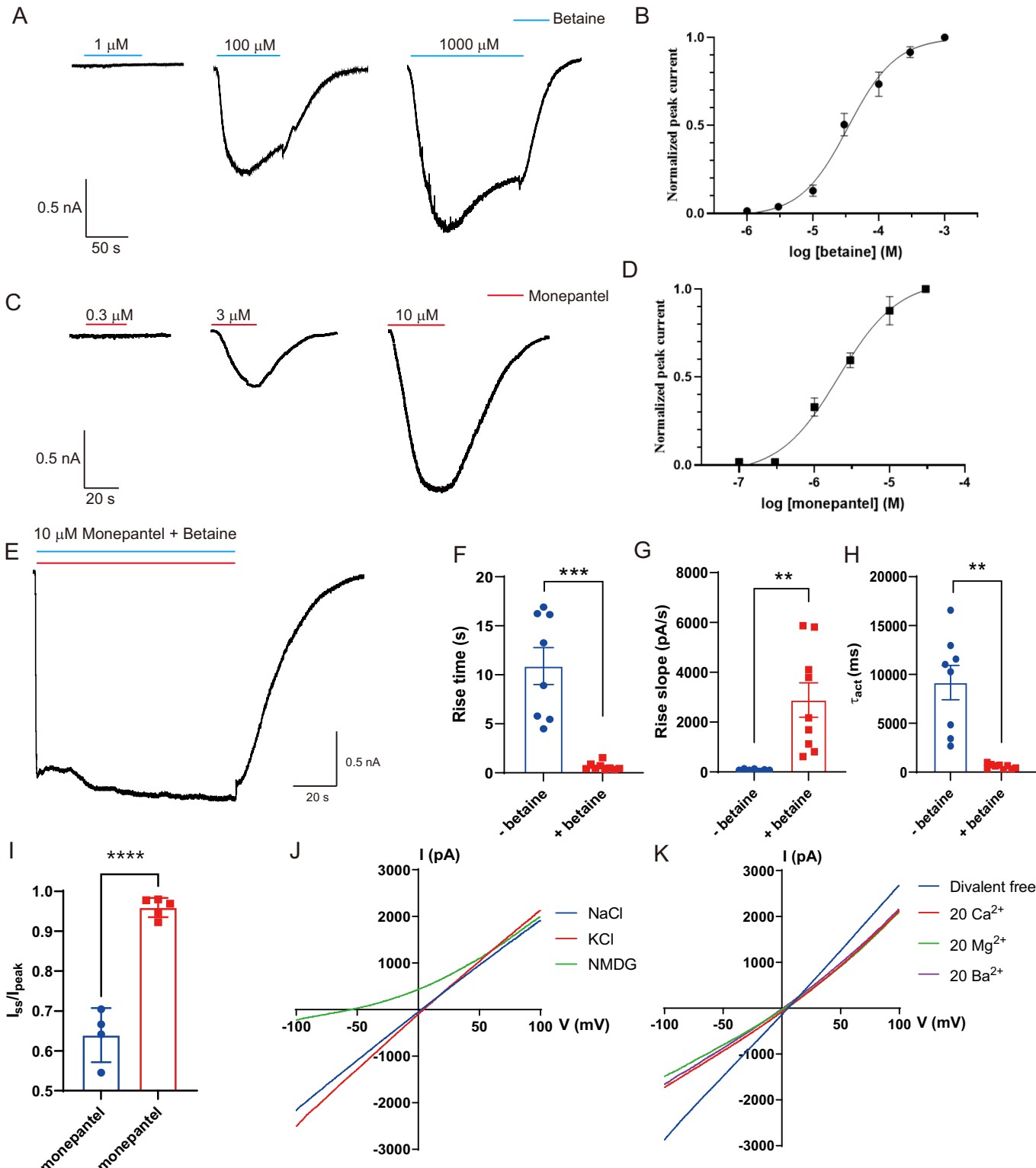

HcMPTL1, also leads to AADs resistance (Rufener et al, 2009). Follow-up electrophysiology studies confirmed that, at higher concentrations, monepantel could act as a super agonist and directly activate ACR-23 or some of its homologs including HcMPTL1 and ACR-20 (Baur et al, 2015; Rufener et al, 2013).

When a lower concentration of monepantel is co-applied with choline, it dramatically potentiates choline elicited currents, not only in ACR-23/HcMPTL1/ACR-20, but also in some other DEG3 family members including *Haemonchus contortus* DEG3/DES2 (Baur et al, 2015; Rufener et al, 2010a; Rufener et al, 2013).

◀  **Figure 1.  Functional characterization of ACR-23 in HEK293 cells.**

(A) Representative whole-cell current evoked by betaine recorded from HEK293 cells overexpressing ACR-23. (B) Dose-dependent activation of ACR-23 by betaine ($n \geq 4$). (C) Representative whole-cell current evoked by monepantel. (D) Dose-dependent activation of ACR-23 by monepantel ($n \geq 4$). (E) Representative ACR-23 current evoked by monepantel with 10 μM betaine. (F) The activation time ($n \geq 8$, $P$ value 0.0009), (G) activation slope ($n \geq 7$, $P$ value 0.0036), and (H) activation time constant ($n = 8$, $P$ value 0.0018) of ACR-23 current evoked by monepantel with or without betaine. (I) The statistics of steady-state current relative to the peak response evoked by agonists ($n \geq 4$, $P$ value $2.25 \times 10^{-5}$). (J) Representative I–V curves of monepantel-evoked current in bath solution containing $Na^+$, $K^+$ or NMDG show that ACR-23 is a monovalent cation channel ($n \geq 3$). (K) Representative I–V curves of monepantel-evoked current in divalent cation-containing bath solution. The reversal potential of monepantel-evoked currents was not notably altered when the extracellular divalent cation concentration was increased. Data represent the mean ± s.e.m. (biological replicates $n \geq 3$). Statistical significance was determined using unpaired t-test with Welch's correction (F–H) or unpaired t-test (I); **$P < 0.01$, ***$P < 0.001$, ****$P < 0.0001$. Source data are available online for this figure.

In addition to monepantel, betaine, a natural compound in seaweed extract which was traditionally used by farmers to prevent nematode infestations, also exerts its nematode-killing activity by over-activating ACR-23 (Peden et al, 2013) and thereby arresting larval development (Langowski et al, 2021). While betaine represents one of the natural compounds used to control parasitic nematodes long before the emergence of synthetic anthelmintics, it is also a compound that exists broadly in animal species, and was recently found to be a possible endogenous ligand of ACR-23 in *Caenorhabditis elegans* (Peden et al, 2013). More recently, researchers have found that betaine is synthesized and released from neurons, and it regulates food search behaviors by targeting the Cys-loop family chloride channel LGC-41 in *Caenorhabditis elegans* (Hardege et al, 2022). In mammals, the transporter BGT-1 is capable of transporting betaine as well as GABA, and betaine has been shown to have anti-epileptic properties (Knight et al, 2017). In addition, as a modified amino acid consisting of glycine with three methyl groups, betaine could function as an osmolyte and methyl donor in vivo (Craig, 2004).

In this study, we determined the cryo-electron microscopy structures of ACR-23 in apo, betaine-bound, and betaine&monepantel-bound states, with bound betaine and monepantel molecules unambiguously modeled. These structural studies, combined with site-directed mutagenesis, electrophysiology, and in vivo locomotion assays, provide valuable insights into the mechanisms of action by betaine and monepantel, which paves the way for future structure-based drug design targeting ACR-23.

## Results

### Functional characterization of ACR-23

The coding sequence of ACR-23 was amplified from cDNA of *C. elegans*, and GFP was fused to its C-terminus. The chimera was then transiently expressed in HEK293 cells for electrophysiology recordings. ACR-23 could be activated by betaine in a dose-dependent manner, with a half-maximal concentration ($EC_{50}$) of ~34.43 μM (Fig. 1A,B), which is largely different from that of previous two-electrode voltage clamp measurement conducted on oocytes (~1 mM), along with a much slower activation kinetics and less desensitization (Peden et al, 2013; Rufener et al, 2013). These differences are likely due to difference in the system used for recording (oocytes vs HEK293 cells), which may have different post-translational modifications or contain distinct modulatory proteins. Monepantel was reported as a super agonist of ACR-23 (Peden et al, 2013; Rufener et al, 2013). In our system, monepantel

directly activated ACR-23, with an $EC_{50}$ value of 2.13 μM (Fig. 1C,D). Moreover, a low dose of betaine (10 μM) was able to boost the activation of ACR-23 by monepantel (Peden et al, 2013). Betaine&monepantel evoked ACR-23 currents barely desensitized, with a greater current amplitude and a faster activation time (Fig. 1E–I). The fact that betaine and monepantel activate ACR-23 cooperatively implies that these two molecules occupy distinct binding sites and have different modes of action, which is well supported by our structural studies, as will be described in later sections.

ACR-23 was reported to be a nonselective cation channel (Peden et al, 2013; Rufener et al, 2013). Consistent with previous studies, the monepantel evoked ACR-23 current has a reversal potential of ~+3.6 mV in our electrophysiological experiments (Fig. 1J). Replacing $Na^+$ in the extracellular solution with an equal amount of $K^+$ did not significantly alter monepantel-induced current, suggesting that ACR-23 is equally permeable to $Na^+$ and $K^+$ (Fig. 1J). However, when extracellular $Na^+$ was replaced by N-methyl-d-glucamine (NMDG), an impermeable organic monovalent cation, the inward current was almost completely abolished (Fig. 1J). To test the divalent cation permeability of ACR-23, we applied 20 mM divalent cation ($Ca^{2+}$, $Mg^{2+}$, or $Ba^{2+}$) in the extracellular solution. Divalent cation decreased the current amplitude and failed to notably shift the reversal potential (Fig. 1K). Thus, ACR-23 is not permeable to divalent cations such as $Ca^{2+}$, $Mg^{2+}$, or $Ba^{2+}$. Collectively, these experiments confirmed that ACR-23 is a nonselective monovalent cation channel.

### Structure determination and overall structure

Inspired by the construct design strategy for α7 nicotinic acetylcholine receptor (Noviello et al, 2021), the loop between M3 and M4 of ACR-23 (residues Lys 386 to Gln 437) was replaced with thermostabilized cytochrome b562 (bRIL) (Chun et al, 2012) to facilitate protein expression. This engineered ACR-23 was then co-expressed with RIC-3 (a protein known to facilitate folding of nicotinic acetylcholine receptors in *c. elegans* (Halevi et al, 2002)) in SF9 cells using a baculovirus-based system (Appendix Fig. S1). The engineered ACR-23 construct has similar electrophysiology behavior as the wild-type channel (Appendix Fig. S2). Therefore, it is a valid target for structural studies. Protein purified in N-dodecyl β-D-maltoside (DDM) supplemented with cholesteryl hemisuccinate (CHS) was subjected to single particle cryo-EM analysis with or without the presence of betaine, or both betaine and monepantel, which yielded three ACR-23 structures in apo, betaine-bound, and betaine&monepantel-bound states at resolution of 2.61, 2.96, and 2.64 Å, respectively (Appendix Figs. S3–5). High quality cryo-EM

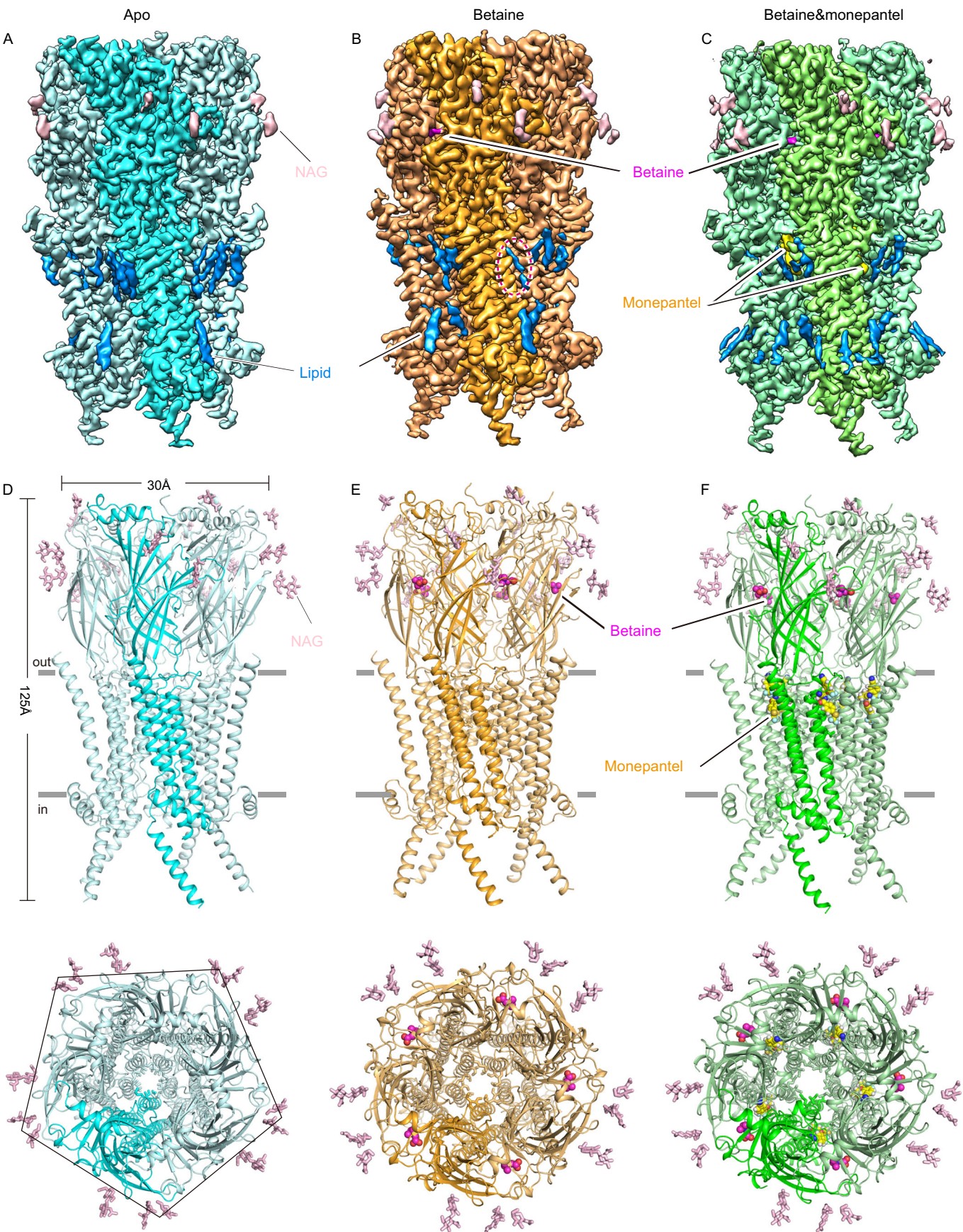

**Figure 2.   Structures of ACR-23 in different states.**

Electron density maps (**A–C**) and cartoon representations (**D–F**) of ACR-23 in apo, betaine-bound, and betaine&monepantel-bound states, as labeled on top. In (**D–F**), side and top views are shown for each structure, at the upper and lower half of each panel, respectively. For all panels, one subunit is highlighted with deeper color. Betaine molecules are shown as magenta surfaces or spheres, whereas monepantel molecules are shown as yellow surfaces or spheres. NAG molecules are shown as pink surfaces or sticks. Bound lipid molecules are shown as blue surfaces in (**A–C**), with the one that partially occupies the monepantel site in betaine-bound state highlighted by a dashed ellipse.

density maps allowed accurate modeling of major parts of the protein except for the intracellular loop between M3 and M4. In addition, 5 glycans that are attached to 3 Asn residues (Asn 53, Asn 97, and Asn 228), as well as bound betaine and monepantel were modeled (Appendix Fig. S6). Lipid-like densities were also observed in our maps (Fig. 2A–C), but were not modeled due to their unknown identity. Notably, one such density partially occupies the monepantel site in betaine-bound state but not in apo and betaine&monepantel-bound states (Fig. 2A–C). These structures are similar to each other except for the local structural changes induced by betaine or monepantel binding (Fig. 2).

ACR-23 adopts a pentameric architecture as other Cys-loop family LGICs (Figs. 2D–F and EV1A) (Gharpure et al, 2020), with each protomer containing a large extracellular domain (ECD) composed almost exclusively of β strands, a transmembrane domain (TMD) with 4 TM helices and a largely disordered intracellular domain (ICD), in which only the helix following the MX helix (named MB herein) is resolved (Figs. 2D–F and EV1B,C). Blast analysis reveals that the closest mammalian homolog of ACR-23 is α7 nicotinic acetylcholine receptor, which is one of the main targets of snake toxins and whose structures have recently been reported (Noviello et al, 2021; Zhao et al, 2021). Despite low sequence identity (~30%), the structure of ACR-23 and human α7 nicotinic acetylcholine receptor are very similar to each other (Fig. EV1B,C), as indicated by a low root mean squared deviation (RMSD) when their apo state structures were superposed (1.9 Å over 1650 residues). The most striking difference lies at the C-terminus of M4. In ACR-23, the whole M4 adopts a straight, continuous helix, whereas in α7 nicotinic acetylcholine receptor, the C-terminus of M4 deviates by ~90° away from other part of M4, forming the so-called "latch helix" (Noviello et al, 2021) (Fig. EV1B,C).

## Ion conduction pathway

A continuous ion conduction pathway is formed along the 5-fold symmetry axis, spanning both ECD and TMD (Fig. EV2A–D). In the ECD, 6 negatively charged residues from each subunit (Asp 106, Asp 107, Asp 112, Glu 120, Asp 122, and Asp 123) point their side chains into the extracellular vestibule and are therefore involved in forming multiple layers of strongly electronegative rings (Fig. EV2A,B), which likely function to attract cations, and is consistent with the fact that ACR-23 is a cation channel (Fig. 1) (Peden et al, 2013; Rufener et al, 2013). Notably, mutation of Asp 112 (D112N) is among the ACR-23 mutations that render *C. elegans* resistant to monepantel (Kaminsky et al, 2008a), implying their important role in the ion conductance of ACR-23. The D112N mutation dramatically reduces the current elicited by monepantel (Rufener et al, 2013). As well, residues in the extracellular vestibule have been implicated in affecting ion selectivity and conductance in

other Cys-loop family LGICs (Hansen et al, 2008; Imoto et al, 1988). In the human α7 nicotinic acetylcholine receptor, 5 copies of Asp 97 in extracellular vestibule form a similar electronegative ring, which has been shown to regulate calcium permeation by forming a constriction in the open state (Noviello et al, 2021). However, there are no such conformational changes in the ECD in our ACR-23 structures.

In the TMD, multiple residues from M2 helices of all 5 subunits line the ion conduction pore, including Glu -1' (Glu 276), Thr 2' (Thr 279), Thr 6' (Thr 283), Leu 9' (Leu 286), Ile 13' (Ile 290), Met 17' (Met 294) and Asp 20' (Asp 297), which are well-defined pore lining residues shared among nAChRs (Gharpure et al, 2020) (Figs. 3A and EV2E,F). Asp 297, which is positioned at the entrance of the ion conduction pore, is an extension of the electronegative extracellular vestibule and likely further attracts cations into the pore (Fig. 3A). In the apo state, multiple constrictions could be observed along the ion conduction pathway at the extracellular half, with the smallest radius of ~1.9 Å, which is too narrow for $Na^+$ or $K^+$ passage, as supported by MD simulations (Fig. 3A–D). Formation of these constrictions involves three residues on M2, 17'Met (Met 294), 13'Ile (Ile 290), and 9'Leu (Leu 286), respectively, which are located in upper (Met 294 and Ile 290) or middle (Leu 286) of the transmembrane pore (Fig. 3A). In the betaine-bound state, these constrictions become wider, with the smallest radius of ~2.3 Å, which is still not wide enough for $Na^+$ or $K^+$ passage, as supported by MD simulations (Fig. 3A–D). Our betaine-bound ACR-23 structure is unlikely in desensitized state, since the desensitization gate is not closed as observed in other LGICs (Gielen and Corringer, 2018) (Fig. EV2G). Therefore, the functional state of betaine-bound ACR-23 structure could not be reliably assigned for now. In the betaine&monepantel-bound state, these constrictions become much wider, with the smallest radius of ~3.1 Å, which is wide enough for dehydrated $Na^+$ or $K^+$ to pass through (Marcus, 1988), as supported by MD simulations (Fig. 3A–D). Therefore, our betaine&monepantel-bound ACR-23 structure is likely in open state. The conformational changes involved in pore opening in TMD are similar to other LGICs (Figs. 3E and EV2H), and will soon be discussed in detail.

## Neurotransmitter pocket and betaine binding induced conformational changes

Like other Cys-loop family LGICs, one neurotransmitter pocket is formed at each of the five interfaces between the neighboring ECDs (Gharpure et al, 2020) (Fig. 2E). Several loops, which are conserved among Cys-loop family LGICs (Gharpure et al, 2020), are involved in the formation of the neurotransmitter binding pocket, including loops A-C from one subunit that forms the 'principal' or (+) face, and loops D-E from the neighboring subunit that forms the

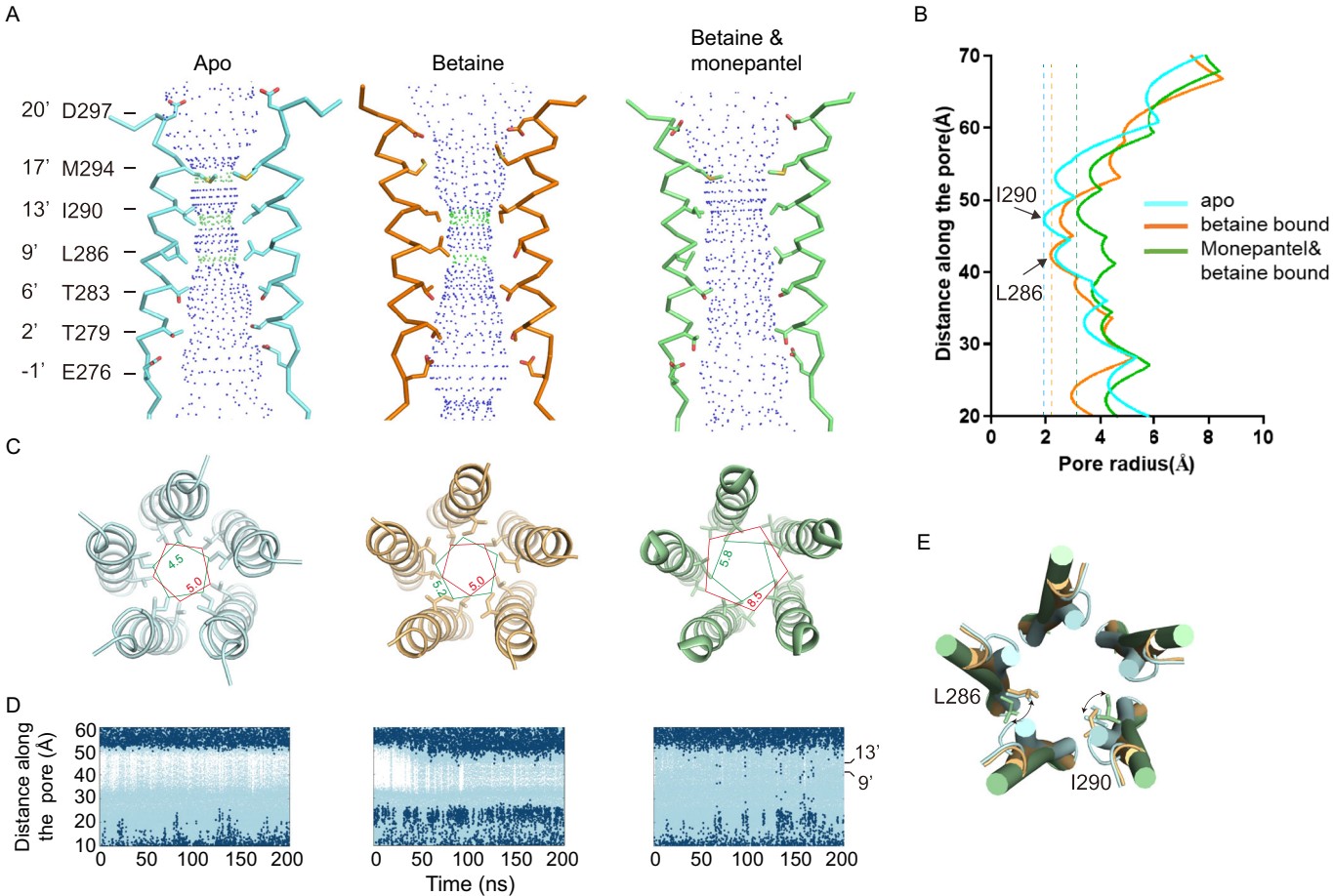

**Figure 3. Ion permeation pathway in the TMD.**

(A) Ion permeation pathway in the TMD, with main chains of M2 and side chains of the pore lining residues from two subunits shown as ribbons and sticks, respectively. (B) Plots of pore radius for TMD of ACR-23 in different states calculated using the HOLE program. Dashed lines indicate the narrowest radius in each state, with contributing residues indicated by arrows and labeled. (C) Top views of pore-lining M2 helices in different states, with key pore-lining residues shown as sticks. Red and green pentagons show changes in side chain position and distances of Leu 286 and Ile 290, respectively. (D) Trajectories of water and $Na^+$ ion coordinates inside the transmembrane pore over 200 ns simulations with a 0.2 V transmembrane potential difference (negative at cytosolic side). Water and $Na^+$ are shown as light blue and deep blue dots respectively, whereas white stretches indicate regions devoid of water and ions. One of three independent 200-ns simulations is shown for each structure. (E) Top view of superposed M2 helices from different states, highlighting conformational changes in the main chains. M2 helices are shown as cylinders, and side chains of key pore-lining residues are shown as sticks. For clarity, side chain of only one key residue is shown on each M2 helix, with curved arrows indicating the direction of their movement. Source data are available online for this figure.

'complementary' or (-) face of the pocket (Figs. 4 and EV1B, and Appendix Fig. S1).

Inspection of cryo-EM densities inside each of the 5 putative neurotransmitter binding pockets of betaine-bound or betaine& monepantel-bound ACR23 structures revealed extra densities that fit well with a betaine molecule, as well as a bridging water molecule (Fig. 4A–C). Betaine, or trimethyglycine, which is the oxidation product of choline, has an overall longish shape similar to choline or its other derivatives (Figs. 4A–C and EV3A–E). To our knowledge, no structures of Cys-loop family LGICs in complex with choline/betaine have been reported previously, hindering a direct structural comparison. However, structural information is available for other choline derivatives, e.g. carbamylcholine, bound to its receptors or acetylcholine-binding protein (AChBP) (Celie et al, 2004; Zarkadas et al, 2022). In addition, structural information is also available for glycine, a betaine metabolite,

bound to human glycine receptor α3 (GlyRα3) (Huang et al, 2017). Bound betaine in ACR-23 is almost parallel to the membrane, markedly different from the bound carbamylcholine in *Lymnaea stagnalis* AChBP (Celie et al, 2004) and *Torpedo californica* nAChR (Zarkadas et al, 2022), which is almost perpendicular to that of betaine in ACR-23 (Fig. EV3B,C,E). On the other hand, the binding pose of betaine in ACR-23 is similar to the binding pose of glycine in GlyRα3 (Huang et al, 2017), except that a water molecule is likely involved in betaine binding in ACR-23, but not in glycine binding in GlyRα3 (Fig. EV3D). Notably, the quaternary amine of betaine and carbamylcholine, and the amine group of glycine, are all caged in an aromatic box (Figs. 4B,C and EV3B–D).

The quaternary amine of betaine is caged in an aromatic box formed by Tyr 116 from loop A(+), Trp 178 from loop B(+), Tyr 222 and Tyr 229 from loop C(+), and Trp 78 from loop D(-), and it engages an archetypical cation-Π interaction with Trp 178, similar

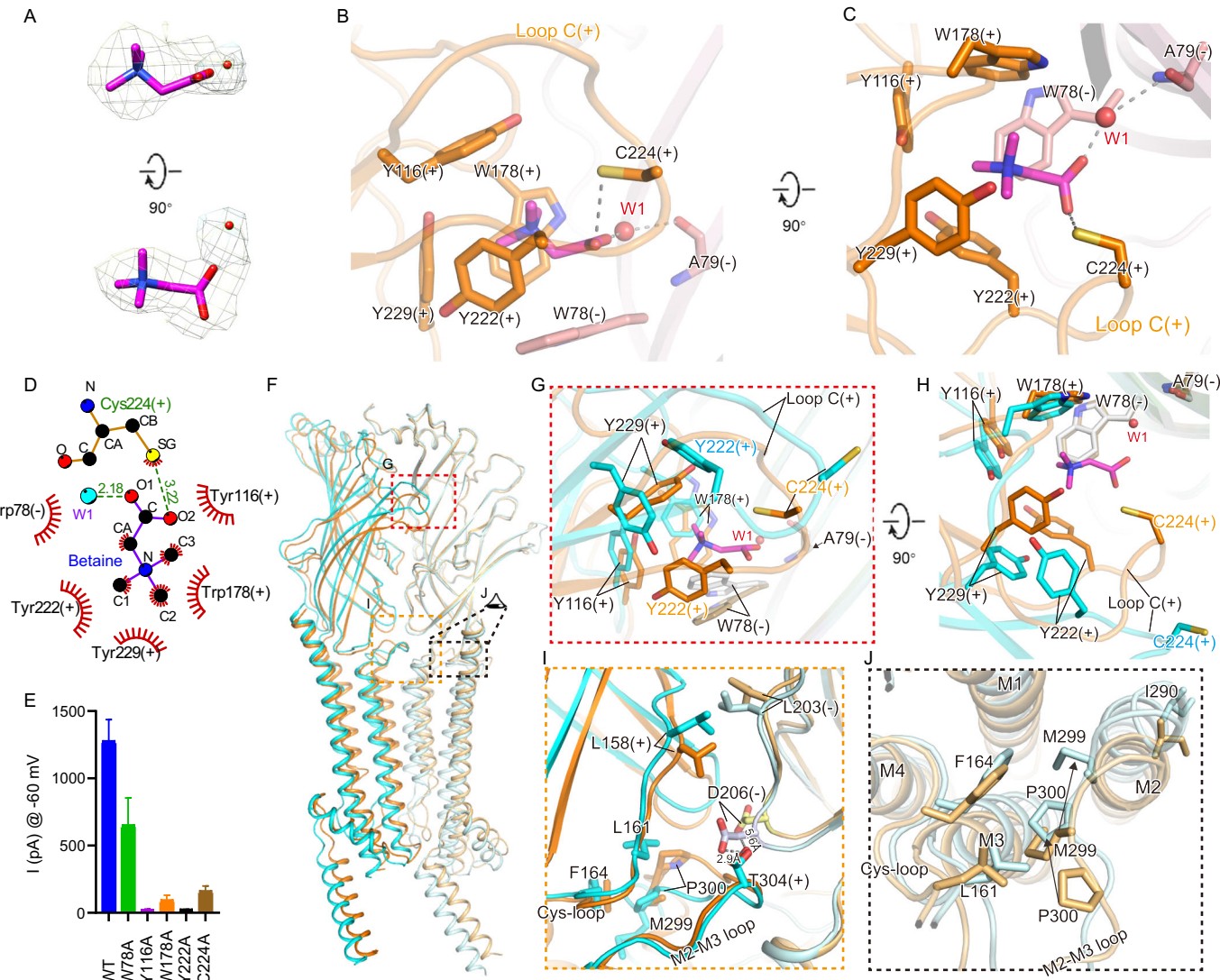

**Figure 4. Neurotransmitter site and betaine binding in ACR-23.**

(A) Electron densities of bound betaine in two views. (B, C), Details of interaction between ACR-23 and betaine in front (B) and top (C) view. (D) Schematic representation of the interactions between betaine and ACR-23 as calculated by LIGPLOT. (E) Peak amplitude of the current for various ACR-23 mutants evoked by 1 mM betaine. Data represent the mean ± s.e.m. (biological replicates n ≥ 3). (F) Overlay of apo and betaine-bound structure of ACR-23, in which the ECD (22–241) of the minus subunit of each structure is used for structure alignment. The plus and minus subunit is colored in different shades of cyan and brown for the apo and betaine-bound structure, respectively. For clarity, only two adjacent subunits are shown. (G, H), Expanded view of the conformational changes around the neurotransmitter site in front (G) and top (H) view. (I) Expanded view of the conformational changes at the inter-subunit interface in the ECD of ACR-23 upon betaine binding. (J) Expanded view of the conformational changes at the ECD-TMD junction upon betaine binding. For (G), (I), and (J), the corresponding regions in (F) are indicated with colored boxes and labeled at the corner. Source data are available online for this figure.

to carbamylcholine binding in *Torpedo californica* nAChR (Zarkadas et al, 2022) (Fig. 4B–D; Appendix Fig. S1). When the residues forming the aromatic box were mutated, e.g., Y116A, W178A, and Y222A, betaine elicited currents were almost completely abolished, whereas for W78A, only ~50% of the current was retained when compared with WT (Fig. 4E). For most of these mutants, the protein expression level and localization are comparable with WT, and could still be activated by monepantel (Appendix Fig. S7). One oxygen atom from the carboxylate group of betaine forms a hydrogen bond with the thiol group of Cys 224 from loop C(+), whereas the other oxygen atom interacts with

carbonyl oxygen of Ala 79 from loop D(-) through a water molecule (Fig. 4B–D). This is consistent with the fact that betaine has a higher potency than choline when activating ACR-23 (Peden et al, 2013), since betaine has an extra carbonyl group, which helps to stabilize its binding to ACR-23 by forming these additional interactions.

While the neurotransmitter pocket in the apo state adopts an open conformation, binding of betaine causes closure of the neurotransmitter pocket, particularly loop C (Fig. 4F–H), and these conformational transitions are similar to other Cys-loop family pLGICs in agonists bound state (Gharpure et al, 2020). There are

also dramatic rearrangements in side chains of the residues involved in betaine binding, particularly Tyr 222 and Tyr 229 (Fig. 4G,H). For most if not all Cys-loop LGICs, disruption of interactions in the inter-subunit interface in the ECD is observed in agonist bound states (Fig. EV3F–H) (Du et al, 2015; Gharpure et al, 2020; Noviello et al, 2021). Similarly, betaine binding in ACR-23 results in disruption or weakening of interactions in the inter-subunit interface, e.g., hydrophobic interaction between Leu 158(+) and Leu 203(-), and the hydrogen bond between Thr 304(+) and Asp 206(-) (Fig. 4I). Situated between the ECD and TMD, the involvement of the ECD-TMD junction of Cys-loop LGICs in ligand induced activation are well documented (Fig. EV3F–H) (Gharpure et al, 2020). By superimposing ECDs of a single subunit of ACR23 in the apo and betaine-bound states, dramatic structural rearrangement is observed in the junction between the ECD and TMD, which are likely essential for propagation of conformational changes from the ECD to TMD following betaine binding (Fig. 4J). In the apo state, Pro 300 from the M2-M3 loop forms Ch/Pi interaction with Phe 164 from the Cys-loop, whereas in betaine/betaine&monepantel-bound states, this interaction is disrupted and Pro 300 slides outward away from the ion conduction pore (Fig. 4J). Consequently, Met 299 in betaine/betaine&monepantel-bound states occupies the position where Pro 300 used to sit in the apo state, and the upper part of M2 is dragged away from the ion conduction pore, leading to a dilated upper pore (Figs. 3 and 4J). Pro 300 are highly conserved among Cys-loop LGICs, and, similar movement of counterparts of Pro 300 is also observed in other Cys-loop LGICs (Fig. EV3F–H). For the TMD, it undergoes rigid body movements upon betaine binding, consistent with the observations in other Cys-loop LGICs (Du et al, 2015; Masiulis et al, 2019) (Appendix Fig. S8).

## Monepantel binding and channel opening

In the cryo-EM density map of the monepantel-bound state, blobs of density that are absent in the apo and betaine-bound state are clearly visible in the inter-subunit interfaces in the TMD, situated at the extracellular half of the membrane (Figs. 2F and 5A). These extra densities match perfectly with monepantel molecules (Fig. 5A). Moreover, the structure of monepantel bound in ACR-23 is highly similar to its crystal structure alone (Ducray et al, 2008) (Appendix Fig. S9). Formation of the monepantel binding pocket involves M2 and M3 from one subunit (referred to as M2(+) and M3(+) hereafter), M1 from the neighboring subunit (referred to as M1(-) hereafter), the M2-M3 loop, as well as the Cys-loop (Fig. 5B). Location of the monepantel binding site is similar to the ivermectin binding site in glutamate-gated chloride channel (GluCl) (Hibbs and Gouaux, 2011), glycine receptor (GlyR) (Du et al, 2015), as well as some allosteric modulators (e.g., diazepam) binding site in the α1β3γ2L GABAA receptor (Masiulis et al, 2019) (Fig. EV4). Therefore, this site seems to be a hotspot for binding of allosteric modulators in Cys-loop LGICs and could be exploited for development of new anthelmintics targeting Cys-loop LGICs in the future.

Monepantel binding in ACR-23 is achieved almost exclusively by hydrophobic interactions, involving Ile 292, Val 295 from M2(+), Ile 309, Ala 310, Tyr 313, Thr 314 from M3(+), Tyr 244, Leu 247, Pro 252 from M1(-), Met 299, Pro 300, Thr 302, Val 306, Pro 307 from the loop between M2(+) and M3(+), and Leu 161 from the cys-loop (Fig. 5B,C). The only exception is Asn 248 from M1(-), which forms hydrogen bond with monepantel (Fig. 5B,C).

Mutation on most of these residues significantly affected the channel activity, which either completely abolished or largely impaired the monepantel-activated ACR-23 currents, except S296G, A310F and T314A (Fig. 5D). For most mutants, the protein expression level and localization are comparable to WT, however, major portion of them could not be activated by betaine (Appendix Fig. S10). The likely reason is, some of these residues are located in the pore-lining M2 and M2-M3 loop, which may affect channel opening and/or propagation of betaine-induced conformational changes towards ion-conducting pore (Fig. 5). Monepantel was developed based on the initial hit AAD 450, with a CF3 group at position 4 of the benzamide moiety (Kaminsky et al, 2008a). Not surprisingly, a considerable portion of the interactions involved in monepantel binding are responsible for coordinating the aminoa-cetonitrile core, which is shared among AADs (Fig. 5A–C; Appendix Fig. S11). Incorporation of $R_1$ and $R_3$ groups on the phenoxy moiety, and addition of a sulfur atom in the $R_4$ group of the benzamide moiety of AAD 450 are essential for optimal efficacy of monepantel (Fig. 5A–C; Appendix Fig. S11). Our structure shows that the $R_1$ group (CN) points upward and fills the void between the Cys-loop and M2-M3 loop, and interacts with residues from these two loops (Fig. 5B,C). The $R_3$ group (CF3) of monepantel points downward towards the intracellular side, and engages hydrogen bonding with Asn 248 and hydrophobic interaction with a few residues, e.g., Leu 247 (Fig. 5B,C). The $R_4$ group (SCF3) of monepantel is positioned in the middle of the membrane and points downward towards the intracellular side, and is coordinated mainly via hydrophobic interactions with Ala 251, Pro 252, and Ile 317 (Fig. 5B,C).

To further verify the binding between ACR-23 and monepantel in vivo, we generated an ACR-23 knock-out C. elegans line (acr-23) by inserting a stop codon in the ACR-23 coding sequence (Appendix Fig. S12A–C). As expected, acr-23 largely lost sensitivity to monepantel, but still retained ~1/3 of monepantel sensitivity of WT, likely due to the existence of other monepantel targets, e.g., other DEG-3 subfamily members such as ACR-20 (Kaminsky et al, 2008a) (Fig. 5E). While complementation of acr-23 with wild type ACR-23 CDS partially restored monepantel sensitivity, complementation with selected mutants, e.g., M299A and Y313A, failed to restore monepantel sensitivity (Fig. 5E), albeit with similar protein expression level as WT (Appendix Fig. S12D). These confirmed the specific interactions between ACR-23 and monepantel in vivo, and the importance of these residues in monepantel coordination and channel activation.

Wedged into the interface between the neighboring subunits, monepantel is sandwiched between M3(+) and M1(-) (Figs. 2F and 5B). Bound monepantel pulls M2(+) outward away from the 5-fold symmetry axis, the extracellular half in particular, and results in a slight rotation in the clockwise direction when viewed from the extracellular side, as manifested by the movement of side chains of Leu 286 and Ile 290 (Figs. 3E, 5F, and EV5). Consequently, side chains of Leu 286 and Ile 290 deviate away from the center of the pore, and eventually cause widening of constrictions at these positions and opening of the ion conduction pore (Figs. 3 and 5F).

# Discussion

Although a wealth of structural information is available for the Cys-loop superfamily of LGICs, particularly for those which are closely

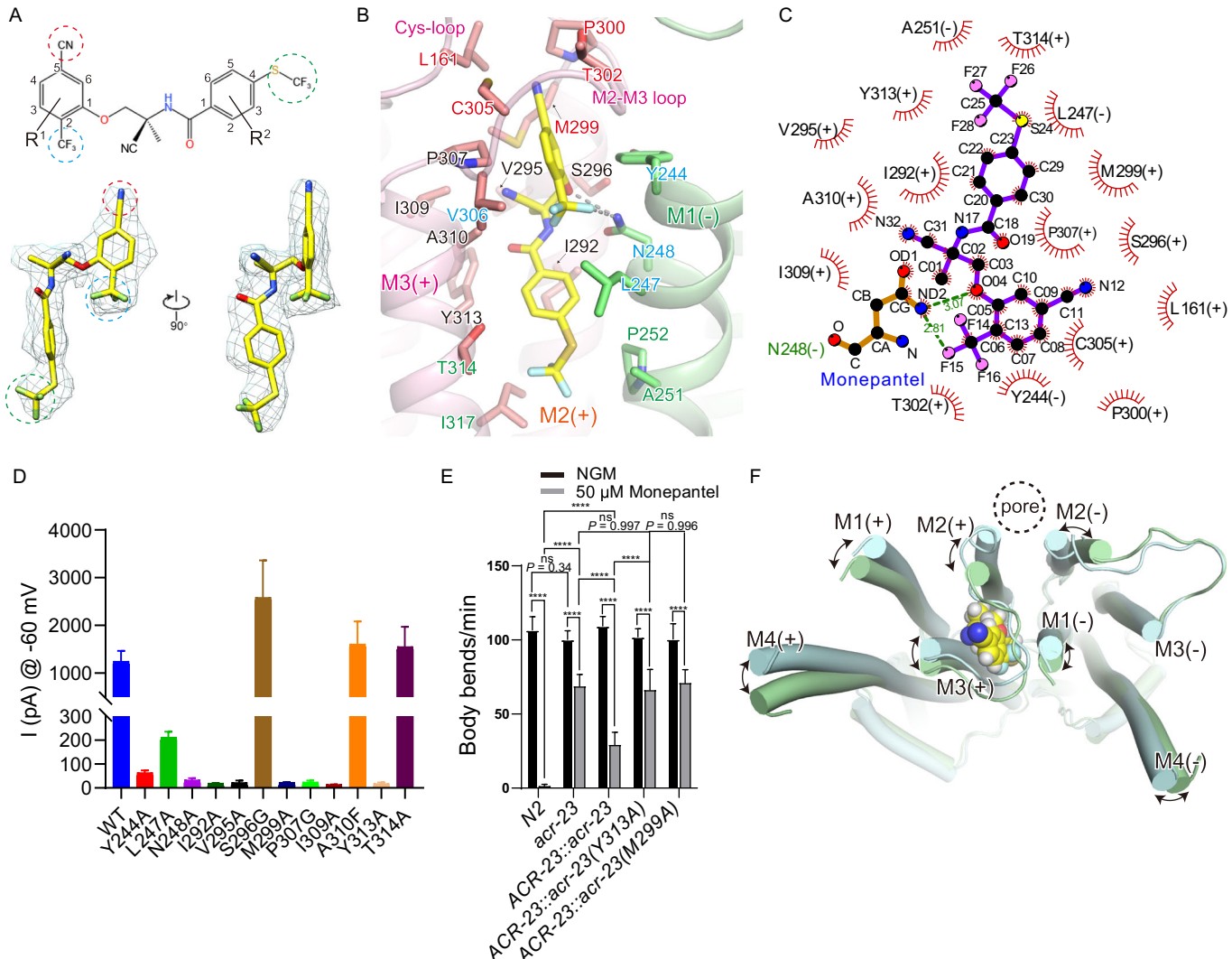

**Figure 5. Allosteric site and monepantel binding.**

(A) Chemical structure of monepantel (top) and electron densities of bound monepantel in two views (bottom). (B) Details of monepantel and ACR-23 interactions. (C) Schematic representation of the interactions between monepantel and ACR-23 as calculated by LIGPLOT. In (A–C), $R_1$, $R_3$, and $R_4$ group in monepantel are circled or labeled in red, blue, or green respectively. (D) Peak amplitude of the current for various ACR-23 mutants evoked by 10 µM monepantel ($n \geq 3$). (E) Rate of body bends of worms pretreated with monepantel on NGM plates in the absence of food ($n \geq 3$). (F) Conformational changes between the TMDs of apo and betaine&monepantel bound ACR-23 structures, with curved arrows indicating their direction of movement. Data represent the mean ± s.e.m. (biological replicates $n \geq 3$). Statistical significance was determined by Tukey's multiple comparison test in two-way analysis of variance (ANOVA); ****$P < 0.0001$ and ns, not significant. Source data are available online for this figure.

related to human diseases (Basak et al, 2018; Du et al, 2015; Hassaine et al, 2014; Hibbs and Gouaux, 2011; Huang et al, 2015; Kim et al, 2020; Laverty et al, 2019; Masiulis et al, 2019; Miller and Aricescu, 2014; Miyazawa et al, 2003; Morales-Perez et al, 2016; Noviello et al, 2021; Nury et al, 2011; Polovinkin et al, 2018; Rahman et al, 2020; Walsh et al, 2018; Yu et al, 2021; Zhu and Gouaux, 2021; Zhu et al, 2018), no structure information is available for the nematode specific DEG-3 subfamily of Cys-loop LGICs. As this subfamily of Cys-loop LGICs are specific to nematodes (Rufener et al, 2010b), they are important and safe targets for development of new anthelmintics. Accurate structural information is crucial for emerging campaigns for structure-based drug development targeting human disease related proteins, which

also holds true for anthelmintics development. In this study, we determined the structures of ACR-23, a member of nematode specific DEG-3 subfamily of Cys-loop LGICs, in different functional states. Our study not only paves the way for development of anthelmintics targeting ACR-23, but also provides a high-quality structure template for other Cys-loop LGICs in the nematode specific DEG-3 subfamily.

Being targeted by both betaine and monepantel, ACR-23 represents an important and relatively new anthelmintics target (Kaminsky et al, 2008a; Peden et al, 2013; Rufener et al, 2013). Overall, our electrophysiology recording results show that two known anthelmintics, betaine, and monepantel, could individually activate ACR-23 (Fig. 1). When they were applied together, a

synergistic effect was observed (Peden et al, 2013) (Fig. 1). Our structural studies unambiguously identified the binding sites and binding poses of betaine and monepantel in ACR-23, and showed that betaine and monepantel bound to ACR-23 at distinct sites and activated the channel with different mechanisms, therefore explaining their synergistic effect. Although binding of betaines in the classical neurotransmitter binding sites was foreseeable given its structural analogy to choline (Peden et al, 2013), there were fewer clues about the binding site of monepantel. The fact that monepantel bound to a pocket in the inter-subunit interface in TMD, a common site for binding of allosteric modulators in Cys-loop family LGICs, encourages future explorations of this site for development of new drugs including anthelmintics targeting Cys-loop LGICs.

In Cys-loop superfamily LGICs, the conserved Cys-loop orchestrats the formation of the ECD-TMD coupling apparatus, which is essential for channel activation by neurotransmitters (Du et al, 2015; Masiulis et al, 2019; Noviello et al, 2021). As human α7 nicotinic acetylcholine receptor shares the highest structure similarity with ACR-23, we compared the structure of the ECD-TMD junction of the α7 nicotinic acetylcholine receptor and ACR-23 (Appendix Fig. S13). Similar to α7 nicotinic acetylcholine receptor and other Cys-loop LGICs, some striking conformational changes were observed at the ECD-TMD coupling region in ACR-23, particularly around FPF motif in the Cys-loop of ECD and Met 299&Pro 300 from the M2-M3 linker of TMD, which are likely essential for channel activation by betaine (Fig. 4J). In addition, structural rearrangements were also observed at the C-terminus of the M4 in the α7 nicotinic acetylcholine receptor, which was called "latch helix" (Noviello et al, 2021) (Appendix Fig. S13). However, in the α7 nicotinic acetylcholine receptor the "latch helix" is bent by ~90° when compared with other parts of M4 helix and is therefore almost parallel to the membrane. By contrast, in all three structures of ACR-23, the counterpart adopts a straight helix (Fig. EV1C). Moreover, residues involved in forming interactions between the "latch helix" and pre-M1 in the α7 nicotinic acetylcholine receptor, e.g., Trp 133, Phe 136, Pro 468 and Phe 470, are poorly conserved in ACR-23 (Appendix Fig. S1).

In the course of monepantel development, 600 derivatives with an aminoacetonitrile core and bearing different aryloxy and aroyl moieties were synthesized and their anthelmintic activity was tested, with the purpose of improving the efficacy of the initial hit (AAD 450) (Ducray et al, 2008; Kaminsky et al, 2008a; Kaminsky et al, 2008b). According to our betaine&monepantel-bound ACR-23 structure, these modifications are necessary for full occupation of the allosteric modulator binding site in ACR-23, and therefore for its optimal efficacy as a super-agonist (Fig. 5; Appendix Fig. S11). In addition, our structures also offer a plausible explanation for some aspects of the Structure-Activity Relationship (SAR) of AADs: (1) While monepantel is the active enantiomer in the racemic mixture of AAD 96, the other enantiomer in the racemic mixture, AAD 96i, is inactive (Kaminsky et al, 2008b) (Appendix Fig. S11). The likely reason is that, in monepantel, the more hydrophobic methyl group points into the interior of the binding pocket and directly interacts with residues on M2, whereas the hydrophilic CN group likely does the same in AAD 96i (Fig. 5; Appendix Fig. S11), and is less efficient in inducing conformational changes in M2, which are essential for channel opening. (2) Position 6 of the phenoxy moiety is very sensitive to substitutions,

in which any group other than H would result in a loss of activity (Ducray et al, 2008). This position is buried in the binding pocket and substitution of this position with other larger groups would probably result in steric hindrance (Fig. 5B).

To identify the targets of AADs, a forward genetic screen for AAD-resistant mutants generated via ethane methyl sulfonate (EMS) mutagenesis was performed (Kaminsky et al, 2008a). 27 independent mutations were identified on ACR-23, including 11 with truncated protein resulting from either premature stop codon or loss of splice acceptor, and the remaining 15 with non-synonymous mutations (Kaminsky et al, 2008a). When mapped onto our ACR-23 structures, 9 out of the 15 non-synonymous mutations are located in the regions that are directly involved in monepantel binding, including M2, M3, the M2-M3 loop and the Cys-loop (Appendix Fig. S14). The 6 other non-synonymous mutations are located in the ECD (Appendix Fig. S14), which is far away from the monepantel binding site. Therefore, most of these mutations probably cause monepantel resistance by affecting the assembly of functional pentameric ACR-23 channels.

In summary, our work provides a comprehensive structural and functional characterization of ACR-23, an important target for anthelmintics. Specifically, we elucidated the mode of action of two anthelmintics, monepantel and betaine, by solving and comparing multiple structures of ACR-23 in different states, which were further validated by mutagenesis, electrophysiology and in vivo behavioral assays. Although full gating cycle of ACR-23 needs further exploration, our study provides valuable information for development of anthelmintics targeting ACR-23 and related channels in the future.

# Methods

## Protein expression and purification

*Caenorhabditis elegans* ACR-23 encoding gene (NCBI refseq NP_001317822) containing a C-terminal strep tag was cloned into a pFastbac vector and heterologously co-expressed with RIC-3 in sf9 cells using the BacMam system. Sf9 cells were grown in SIM SF Expression Medium (Sino Biological Inc), supplemented with 2% fetal bovine serum (FBS, Biological Industries). The baculovirus was generated in Sf9 cells following standard protocols using transfection reagent X-tremeGENE™ (Roche) and used to infect sf9 cells at a ratio of 1:40 (virus: sf9, v:v).

48 h after infection, cells were harvested by centrifugation at $4000 \times g$. Subsequent purification steps were carried out at 4 °C. The cell pellet was resuspended in buffer A (50 mM Tris pH 8.0, 200 mM NaCl), supplemented with 1 mM PMSF and disrupted by sonication on ice. The lysate was centrifuged at 40,000 RPM for 1 h to pellet membranes, which were then Dounce homogenized in buffer A supplemented with 1 mM PMSF. ACR-23 was extracted with 1% (w:v) n-Dodecyl-beta-Maltoside (DDM) supplemented with 0.02% (w:v) Cholesteryl hemisuccinate (CHS) by gentle agitation for 2 h. Insoluble material was removed by centrifugation at 20,000 RPM for 40 min and the supernatant was incubated with Strep-Tactin®XT (IBA Lifesciences) for 2 h with gentle agitation. The resin was then collected in a disposable gravity column (Bio-Rad), washed with buffer A supplemented with 0.05% (w:v) DDM, and finally eluted with elution buffer provided by manufacturer,

supplemented with 0.05% (w:v) DDM and 0.002% (w:v) CHS. The eluant was concentrated, and further purified by size exclusion chromatography on a Superose 6 10/300 GL column (GE Heathcare) pre-equilibrated with buffer B (20 mM Tris pH 8.0, 150 mM NaCl) supplemented with 0.05% (w:v) DDM and 0.002% (w:v) CHS. The peak was collected and concentrated to ~4 mg/ml for cryo-electron microscopy analysis.

## Cryo-EM sample preparation and data acquisition

To obtain the ligand/agonist bound structures, ACR-23 protein was first incubated with 1.5 mM betaine (Solarbio) dissolved in distilled water and/or 5 mM monepantel dissolved in 10% (v:v) DMSO for 30 min on ice before preparation of grids. Procedures for grids preparation for all samples were the same hereafter. In brief, C-flat R 1.2/1.3 holey carbon grids were first glow discharged for 20 s using a Pelco easiGlow glow discharge unit and 3 μL sample was applied to the surface of the grid at temperature of 6 °C and humidity level of 100%. Grids were then blotted for 1.5 s before being plunge-frozen in liquid ethane using Vitrobot Mark IV. The cryo-specimens were loaded onto a Titan Krios transmission electron microscope (Thermo Fisher) operated at 300 kV for data collection. The microscope is equipped with a GIF-Quantum energy filter (Gatan), which was used with a slit width of 20 eV. Automatic data collection was performed using EPU software. Images were recorded with Gatan K2 direct electron detectors operating in super-resolution counting mode at pixel size of 0.84 Å, 0.84 Å, and 0.84 Å, respectively (Appendix Table S1). The exposure was performed with a dose rate of 15 e-/pixel/s and an accumulative dose of ~50 e-/Å$^2$ for each image which was fractionated into 36 movie-frames. The final defocus ranges of the datasets were approximately -(1.2–2.4) μm (Appendix Table S1).

## Image processing and 3D reconstruction

For the apo dataset, the dose-fractionated image stacks were subjected to beam-induced motion correction using MotionCor2 (Zheng et al, 2017). Initial contrast transfer function (CTF) values for each micrograph were calculated with CTFFIND4 (Rohou and Grigorieff, 2015). Micrographs with an estimated resolution limit worse than 5.5 Å were discarded in the initial screening. A set of ~1000 particles were manually-picked and subjected to 2D classification to generate templates for auto-picking against the entire dataset. The subsequent image processing and reconstruction were performed using cryoSPARC (Punjani et al, 2017). 3,400,894 particles were picked from 7399 micrographs. The picked particles were then extracted and subjected to three rounds of reference-free 2D classification in cryoSPARC, which yielded 516,001 particle projections. This subset was subjected to one round of Hetero refinement. The predominant classes containing two subsets of 402,997 best particles shows the clear features of secondary structural elements. These particles were subjected to a non-uniform refinement, which yielded a reconstruction at 2.61 Å resolution (Appendix Fig. S3).

The betaine and betaine&monepantel datasets were processed similarly to the apo dataset. The detailed image processing and reconstruction are shown in Appendix Figs. S4, 5. Local resolution estimate was performed with cryoSPARC (Punjani et al, 2017).

## Model building, refinement, and validation

Model building was manually conducted in Coot (Emsley et al, 2010), guided by Alphafold predicted model (AF-G5EG88-F1). Accurate residue registration was achieved based mainly on the clearly defined densities for bulky residues (Phe, Trp, Tyr, and Arg). Models were refined against cryo-EM maps using real-space refinement in PHENIX (Afonine et al, 2013), with secondary structure and non-crystallography symmetry restraints applied. For modeling and refinement of the monepantel molecule, SMILE strings of these molecules were inputted into eLBOW (Moriarty et al, 2009) implemented in PHENIX to generate a CIF file. The statistic of the models' geometries was generated using MolProbity (Chen et al, 2010). Pore radii were calculated using the HOLE program (Smart et al, 1996). The diagram for protein and ligand interaction was generated by the LIGPLOT program (Wallace et al, 1995). All the figures were prepared in PyMol (Schrodinger 2015) or Chimera (Pettersen et al, 2004). Chemical structures were drawn using KingDraw.

## Electrophysiology

All electrophysiological recordings were obtained using patch-clamp recordings for macroscopic current of ACR-23 and its mutants over-expressed in HEK293 cells. For transfected cells, patch-clamp recordings were conducted 24–48 h after transfection. Patch pipettes were pulled from PC-10 (NARISHIGE) and heat polished to a resistance of 3–10 MΩ. Patch-clamp recordings were performed with an Axopatch-200B amplifier and Axon Digidata 1550 A driven by Clampex10 software (Molecular Devices). The current signals were filtered at 1 kHz and digitized at a 10 kHz sampling frequency. All values were measured using Clampfit 10.6 Software.

To apply monepantel or betaine during patch-clamp recording, a rapid solution changer with a gravity-driven perfusion system was used (RSC-200, Bio-Logic). Each solution was delivered through a separate tube so there was no mixing of solutions. A pipette tip with a membrane patch or a whole-cell patch was placed right in front of the perfusion outlet during recording to ensure the solution exchange was complete.

To test for functionality of ACR-23 or its mutants, cells were held at −60 mV and responses were evoked with monepantel or betaine. The bath solution contained 165 mM NaCl and 10 mM HEPES, pH 7.3. The pipette solution contained 140 mM KCl and 10 mM HEPES, pH 7.2. Data of whole-cell voltage clamp experiments were fitted using the dose–response function. The half-activation concentrations of betaine or monepantel were derived from fits of the dose–response curves to the function:

$$I = A_1 + \frac{A_2 - A_1}{1 + 10^{(\log x_0 - x)P}} \quad (1)$$

where $I$ is the current amplitudes after application of activators; $A_1$ and $A_2$ are constants between 0 and 1; $x$ is the concentration of activator; $x_0$ is the concentration when 50% activation was reached (EC$_{50}$), $P$ is the Hill constant.

To test the ion permeability, the pipette solution contained 165 mM NaCl and 10 HEPES, pH 7.3. The NaCl in the bath solution was replaced with an equal concentration of KCl or NMDG chloride. The whole-cell currents were evoked with 10 μM monepantel and recorded by voltage ramp from −100 mV to +100 mV in 1000 ms.

To calculate the activation time and slope of ACR-23, we measured the rise time and rise slope from 10% to 90% of ACR-23 current using

Clampfit 10.6 Software. To determine the activation kinetic of ACR-23, the activation time constant ($\tau_{\text{activation}}$) of currents was analyzed. We selected the activation phase of ACR-23 fitting to a single exponential function:

$$I = Ae^{-t/\tau} + C \tag{2}$$

where $I$ is the current amplitude, t is the time, $A$ is the amplitude constant, and $\tau$ is the time constant.

## Protein expression level and subcellular localization analysis

To determine the expression level of ACR-23 variants, one dish of cells for each construct were cultured in the same way as for electrophysiology recordings. Pelleted cells were subjected to extraction with 1% DDM, followed by centrifugation (14,000 RPM). Total extract and supernatant were subjected to western blot analysis using antibodies against GFP (Abcam) and GAPDH (ZSGB-BIO), which served as internal standard. For protein localization, plasma membrane marker KRas GTPase in fusion with mCherry (co-transfected with 0.5 μg of plasmid) was co-expressed with the ACR-23. Differential interference contrast (DIC) and fluorescence images were taken by an inverted confocal microscope (LSM800; Carl Zeiss) using a 63×/1.4-NA oil objective lens at 20 °C. Images were processed and analyzed with ZEN 2 blue software (Carl Zeiss). Wavelengths for excitation of GFP and mCherry were 488 and 561 nm, respectively, and those for emission of GFP and mCherry were 470–510 and 560–600 nm, respectively.

## MD simulations

Systems for the ACR-23 structures were built using the CHARMM-GUI server (Jo et al, 2008), with each structure being embedded in a palmitoyl-oleyl-phosphatidylcholine (POPC) bilayer hydrated with water molecules (using TIP3P model). 700 mM NaCl was added to each system. All simulations were carried out with the Gromacs package 2021.4 (Pall et al, 2020) using the CHARMM36m force field (Klauda et al, 2010). Following 50,000 steps of energy minimization with the steepest descent algorithm, a 1125 ps of equilibration at a constant temperature of 300 K was performed, during which positional restraints were gradually reduced, and Berendsen thermostat and barostat were applied. During production runs, temperature and pressure were controlled with a Nose-Hoover thermostat and a Parrinello-Rahman barostat. For channel hydration and permeability analysis, 3 independent replicates of 200 ns long simulations were performed for each system, with positional restraints applied to the protein backbone. Analyses were performed using CHAP (Klesse et al, 2019).

## C. elegans strains, transgenics, behavioral assay, and protein expression detection

C. elegans strains were grown and maintained on NGM medium seeded with E. coli OP50 at 20 °C. The Bristol strain N2 was used as the wild type.

The ACR-23 knockout strain (acr-23) was generated using CRISPR/Cas9 gene editing technology (Arribere et al, 2014). The single-guide RNA (sgRNA) of ACR-23 was predicted to be located

at the first exon (http://crispr.mit.edu), and the targeting sequence (5'-AGGATTGATACCTAAAGTC-3') was cloned into the pDD162 vector (Appendix Table S2), which expresses the Cas9 enzyme. A repair template for ACR-23 was designed to remove the cleavage site. To facilitate the identification of mutants, the repair template contained a restriction enzyme cleavage site (Nhe I), as well as a stop codon and an additional base, which serves to cause frameshift and prevent residual translation. dpy-10 was used as a positive control marker (Smith et al, 2020). The ACR-23 and dyp-10 sgRNA containing pDD162 constructs (20 ng/μL each), and repair templates for ACR-23 and dyp-10 (2 μmol/L each) were co-injected into gonads of young adult animals. Roller and dumpy F1 worms were singled into new NGM plates, and the F2 progeny was examined by PCR amplification and restriction enzyme (Nhe I) digestion. The positive ones were further confirmed by sequencing.

Extrachromosomal transgenes were generated by using standard microinjection methods (Berkowitz et al, 2008). The 2 kb promoter plus the CDS of ACR-23 were ligated into the pPD49.26 vector, which has a mCherry tag. The resulting construct and odr-1p::GFP marker were co-injected into acr-23 worms.

For behavioral assays in the presence of monepantel, we seeded NGM plates with E. coli strain OP50 mixed with 50 μM monepantel. These plates were allowed to dry at room temperature. For each line, 20 worms were used for behavioral assays. In brief, L4 larvae were transferred to OP50 and monepantel-containing plates from NGM plates. 24 h later, worms were picked, one at a time, and soaked in a drop of M9 buffer placed in the center of a food-free NGM plate. The worms were allowed to acclimate for 1 min and the number of body bends in the following 1 min was counted. Each swing from head to tail along the longitudinal axis was considered as one count.

For protein expression detection, a 6 cm dish of worms for each line were soaked in RIPA lysis buffer and homogenized by sonication. After boiling, lysate was subjected to western blot analysis using antibody against mCherry (Proteintech). β-tubulin was used as internal standard and was detected using its antibody (Affinity).

## Statistics

Statistical analyses, e.g., two-tailed Student's t-test and two-way analysis of variance (ANOVA) followed by Tukey's test, were conducted in this study. All statistical data are presented as mean ± standard error of mean (s.e.m.). Experiments were independently repeated at least three times with similar results. The mean values of the control and experimental groups were compared for significant differences. $P$ values below 0.05 were considered statistically significant.

# Data availability

Structure coordinates and cryo-EM density maps have been deposited at the Protein Data Bank and Electron Microscopy Data Bank under accession numbers 8ZFL and EMD-60064 for ACR-23 in apo state; 8ZFM and EMD-60065 for ACR-23 in betaine-bound state; and 8ZFK and EMD-60063 for ACR-23 in betaine&mone-pantel-bound state. Other structure coordinates analyzed in this paper are indicated in the text.

The source data of this paper are collected in the following database record: biostudies:S-SCDT-10_1038-S44318-024-00165-7.

## Peer review information

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

## Acknowledgements

We thank Dr. Bin Qi for suggestions and Dr. Leonard Krall for proofreading the manuscript. This work was supported in part by grants from the Ministry of Science and Technology of China (2020YFA0908503 to QC), the National Natural Science Foundation of China (32071202 and 32271012 to QC; 32100129 to SL; 31371066 and 32371005 to YL), Tianjin Fund for Distinguished Young Scholars (20JCJQJC00080 to QC), Yunnan Fund for Distinguished Young Scholars (202401AV070004 to QC), the Xingdian Scholar Fund of Yunnan to QC, and the Shanghai Municipal Science and Technology Major Project (No. 184319071000 and 19140903102 to YL), the Postgraduate Research and Innovation Foundation of Yunnan University (KC-23236410 to FL).

## Author contributions

**Fenglian Liu**: Data curation; Formal analysis; Investigation; Writing—review and editing. **Tianyu Li**: Data curation; Investigation; Writing—review and editing. **Huihui Gong**: Data curation; Investigation. **Fei Tian**: Investigation. **Yan Bai**: Investigation. **Haowei Wang**: Investigation. **Chonglin Yang**: Conceptualization. **Yang Li**: Data curation; Formal analysis; Funding acquisition; Visualization; Writing—review and editing. **Fei Guo**: Conceptualization; Data curation; Supervision; Writing—review and editing. **Sheng Liu**: Conceptualization; Data curation; Formal analysis; Investigation; Writing—original draft. **Qingfeng Chen**: Conceptualization; Data curation; Supervision; Funding acquisition; Writing—original draft; Project administration; Writing—review and editing.

Source data underlying figure panels in this paper may have individual authorship assigned. Where available, figure panel/source data authorship is listed in the following database record: biostudies:S-SCDT-10_1038-S44318-024-00165-7.

## Disclosure and competing interests statement

The authors declare no competing interests.

# Expanded View Figures

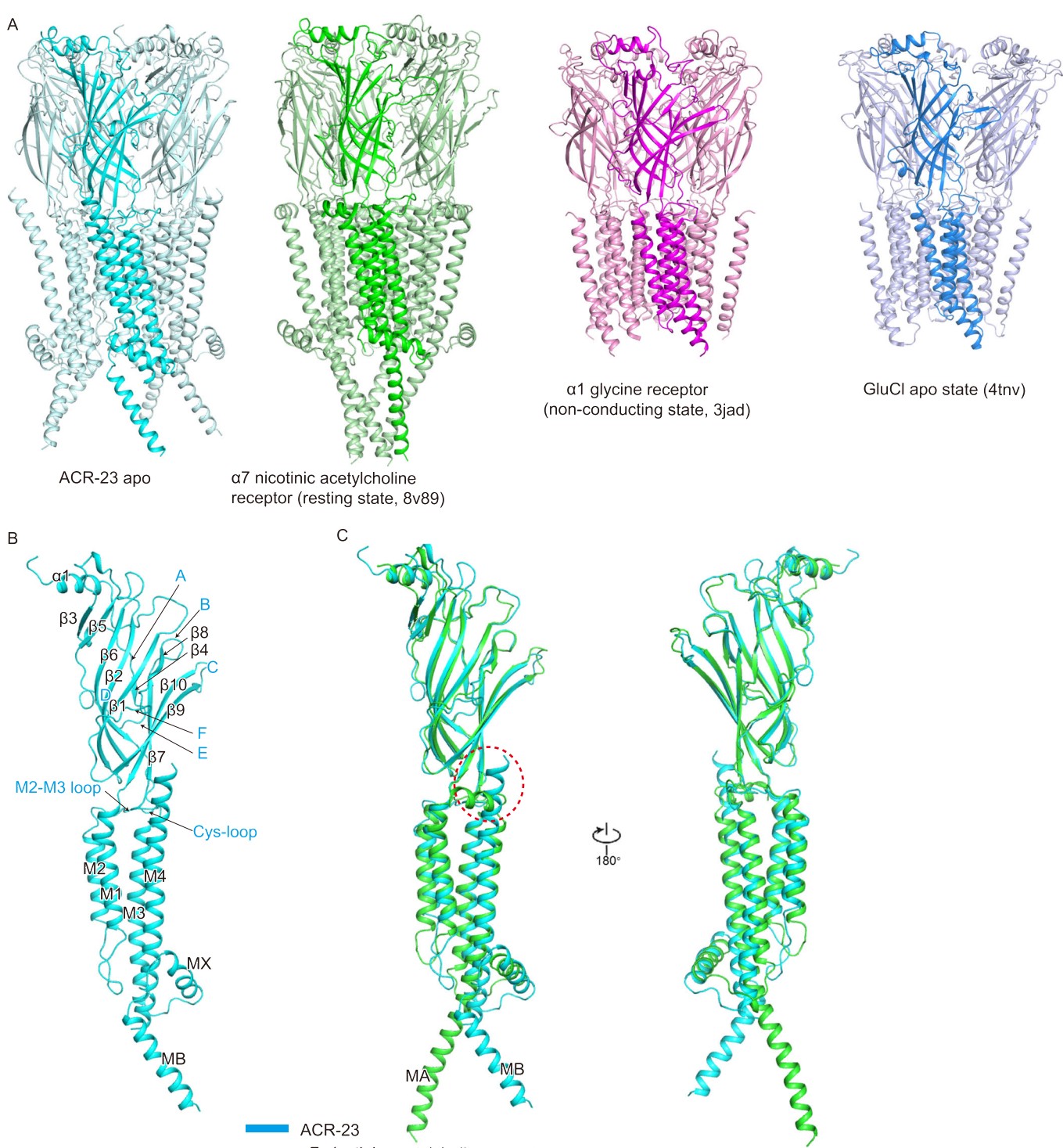

**Figure EV1. Comparison of ACR-23 structure with selected Cys-loop LGICs.**

(A) Comparison of apo structure of ACR-23 and representative Cys-loop LGICs. (B, C) Comparison of protomer structure of ACR-23 and α7 nicotinic acetylcholine receptor. In (B), each secondary-structure element is labeled in protomer structure of ACR-23 (shown as cartoon representations). In (C), superimposition of ACR-23 and α7 nicotinic acetylcholine receptor protomer structure is shown in two views, with the red circle highlighting the difference in the C-terminus of M4 (latch turn and latch helix in α7 receptor). Apo structure of ACR-23 and α7 nicotinic acetylcholine receptor structure in resting state (pdb# 7koo) are used for comparison, and are colored cyan and green respectively.

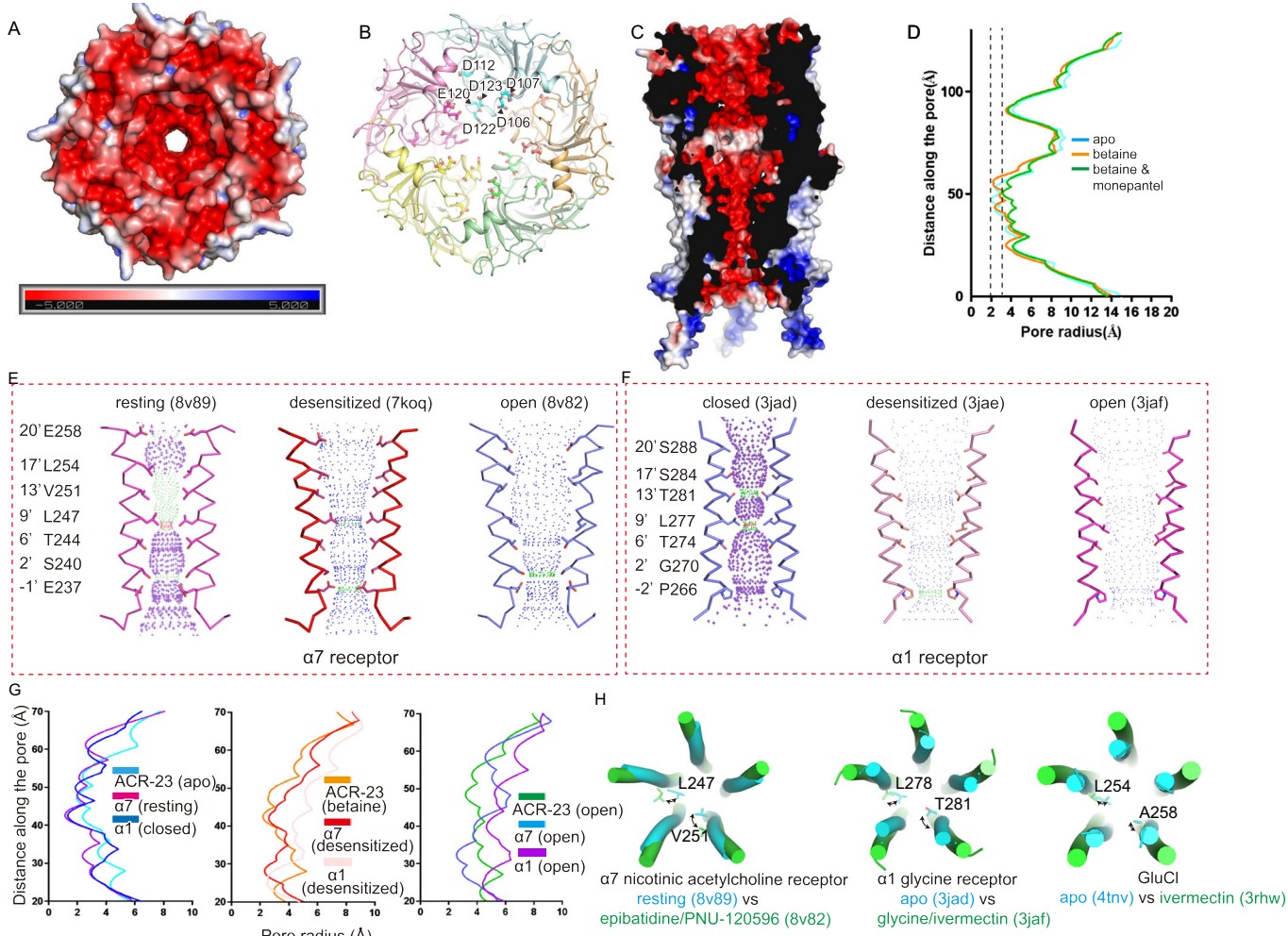

**Figure EV2. Comparison of ion conduction pores of ACR-23 with selected LGICs.**

(**A**) Surface electrostatic potential of ACR-23 viewed from top. (**B**) Top view of the ECD, with main chains shown as cartoons and each subunit colored differently. Rings of negatively charged residues are shown as sticks, and only one subunit is labeled for clarity. (**C**) Surface electrostatic potential along the ion permeation pathway in ACR-23, highlighting its electronegative nature. (**D**) Pore radii along the ion permeation pathway in different states of ACR-23 as calculated by the HOLE program. (**E**, **F**) Ion permeation pathway in the TMD of α7 (**E**) and α1 (**F**) receptors in closed, desensitized and open states, with main chains of M2 from two subunits and the pore lining residues shown as ribbons and sticks respectively. (**G**) Plots of pore radius for TMD of α7 and α1 receptors in closed, desensitized and open states calculated using the HOLE program, and their comparison with ACR-23 apo, betaine-bound and betaine&monepantel bound states. (**H**) Top view of superposed M2 helices from closed and open states of α7, α1, and GluCl receptors, highlighting conformational changes in the main chain and movements of the pore-lining residues. M2 helices are shown as cylinders, and side chains of key pore-lining residues are shown as sticks. For clarity, side chain of one key residue is shown on each M2 helix, with curved arrows indicating the direction of their movement.

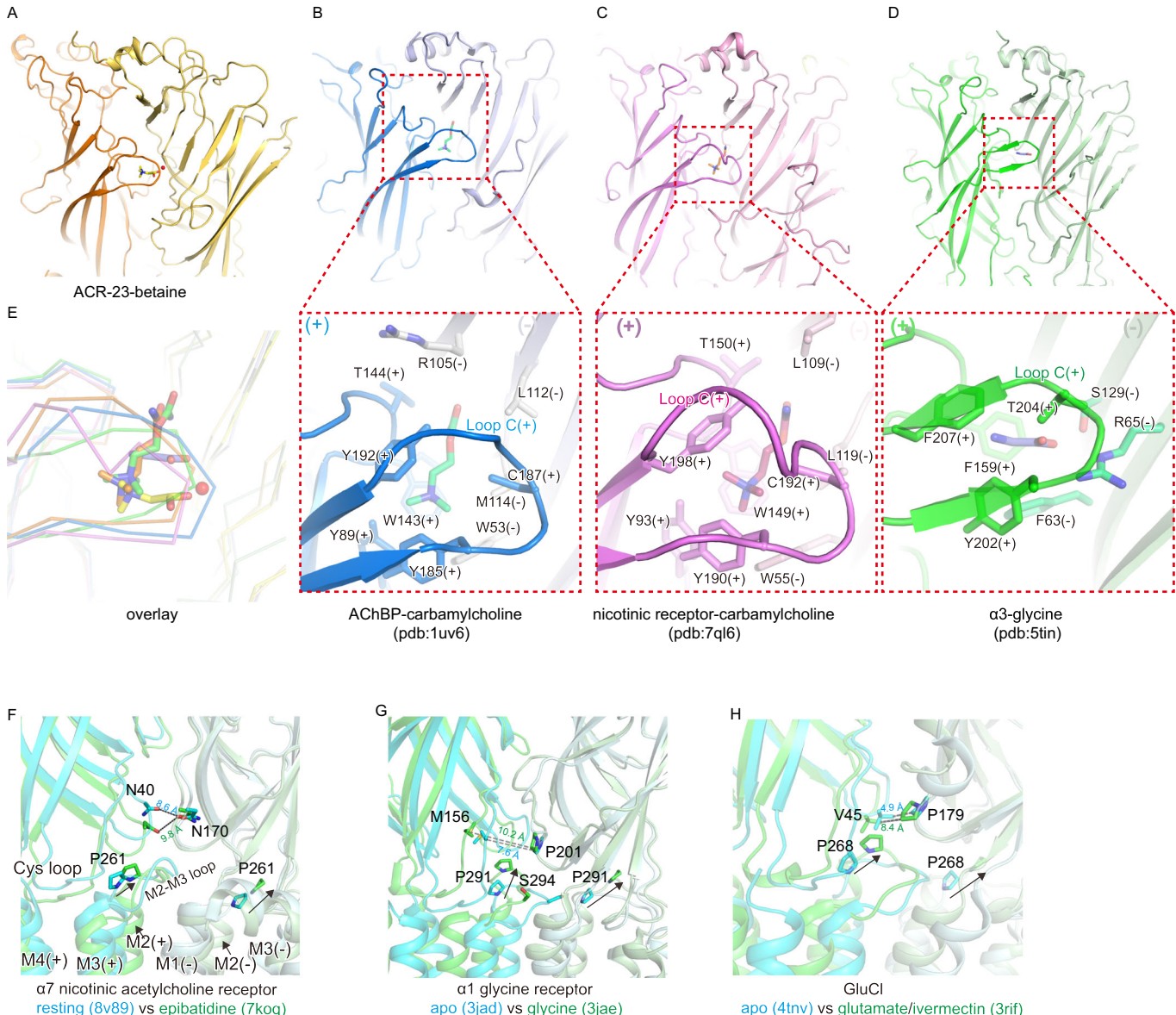

**Figure EV3. Comparison of neurotransmitter binding sites among selected LGICs and conformational changes in the ECD-TMD junction induced by neurotransmitter binding.**

(A–D) Two subunits of selected LGICs are shown as cartoons and colored differently, whereas neurotransmitters are shown as sticks. The name, functional state and pdb accession number of each LGIC are indicated at the bottom of each panel. In (C, D), expanded views of neurotransmitter binding sites of selected LGICs are shown in dashed boxes, with key residues shown as sticks. (E) Overlay of bound neurotransmitters in selected LGICs, highlighting the difference in their binding poses. (F–H) Overlay of closed and neurotransmitter-bound structure of selected LGICs, in which the ECD of the minus subunit of each structure is used for structure alignment. The plus and minus subunit is colored in different shades of cyan and green for the apo and neurotransmitter-bound structure, respectively. For clarity, only two adjacent subunits are shown.

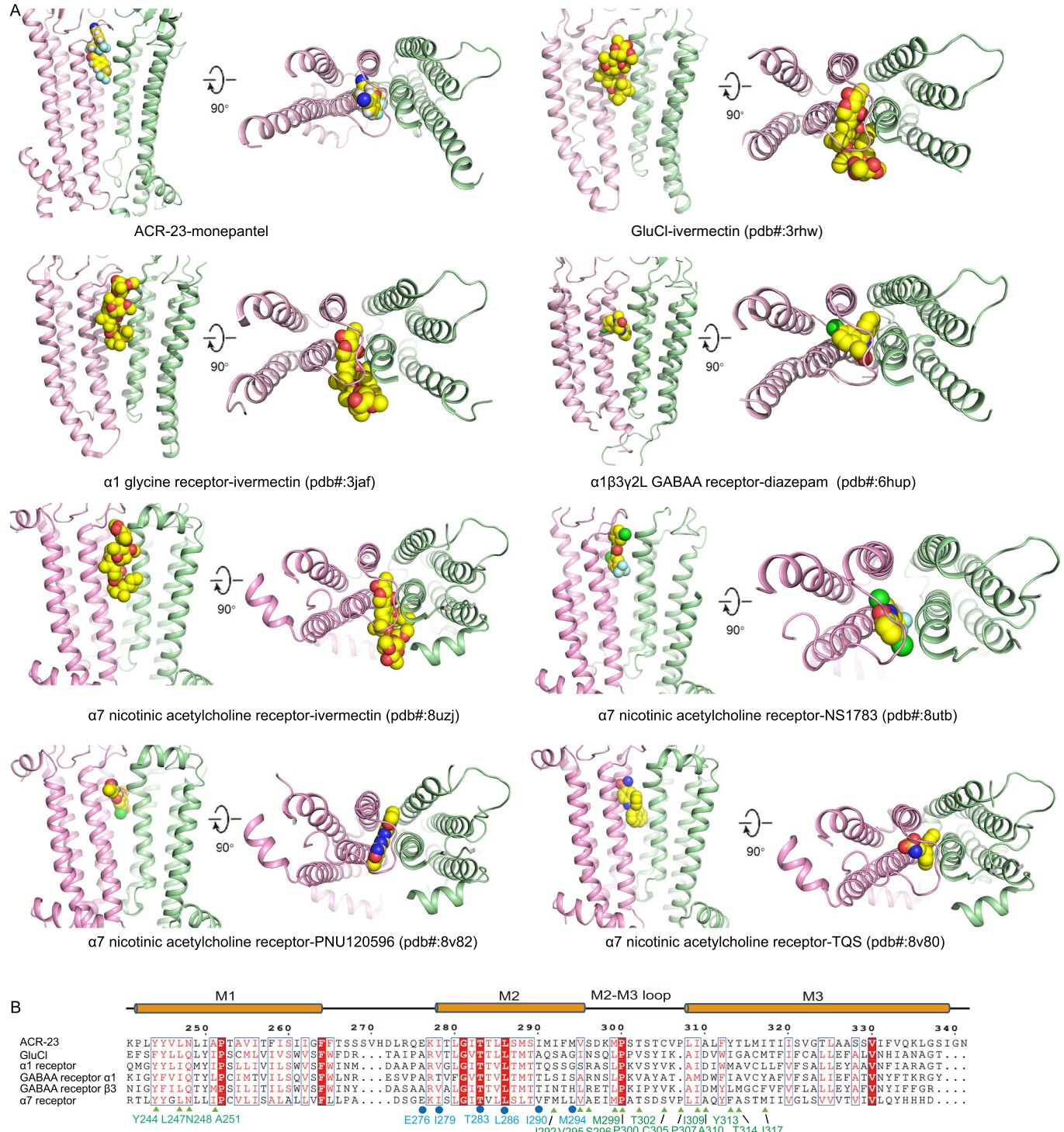

**Figure EV4. Comparison of allosteric modulator binding sites among selected LGICs.**

(**A**) Comparison of binding poses of allosteric modulator among selected LGICs. For each structure, two subunits are shown as cartoons and colored differently, whereas the bound allosteric modulators are shown as spheres. (**B**) Local sequence alignment spanning M1-M3 of ACR-23 and selected LGICs. Green triangles mark residues involved in monepantel binding, whereas blue circles mark pore-lining residues.

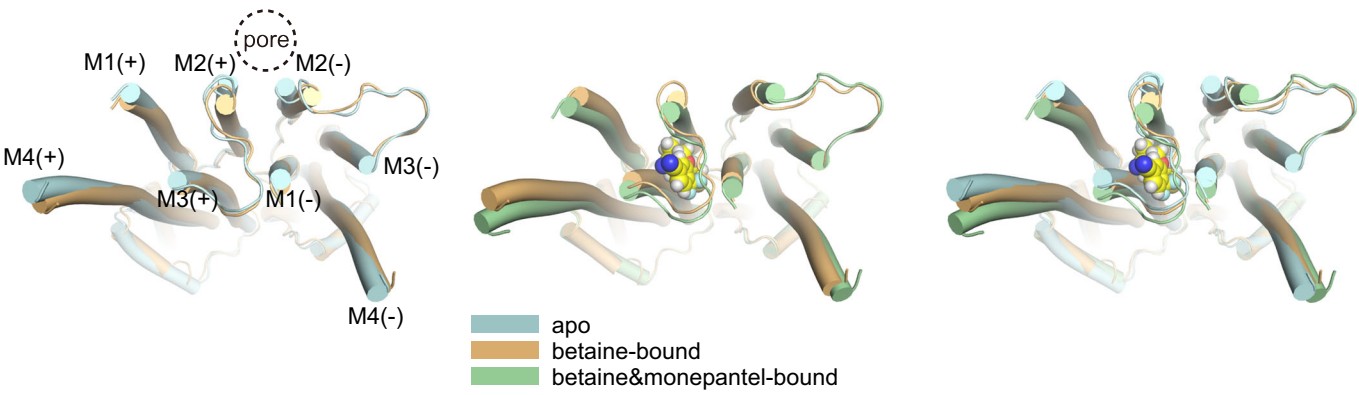

**Figure EV5.  Conformational changes between TMDs of ACR-23 structures in different states.**

Curved arrows indicate the direction of movement. Only two subunits are shown for clarity. Main chains are shown as cartoons and bound monepantels are shown as spheres.

