## [Peer Review File · The EMBO Journal]

Structural insights into the molecular effects of the anthelmintics monepantel and betaine on the *Caenorhabditis elegans* acetylcholine receptor ACR-23

Qingfeng Chen, Fenglian Liu, Tianyu Li, Huihui Gong, Fei Tian, Yan Bai, Haowei Wang, Chonglin Yang, Yang Li, Fei Guo, and Sheng Liu

Corresponding author(s): Qingfeng Chen (qingfengchen@ynu.edu.cn) , Fei Guo (guofei@simm.ac.cn), Sheng Liu (sliu520@mail.ustc.edu.cn)

Review Timeline:

Submission Date:	24th Jan 24
Editorial Decision:	26th Feb 24
Revision Received:	27th May 24
Editorial Decision:	14th Jun 24
Revision Received:	27th Jun 24
Accepted:	28th Jun 24

Editor: Ioannis Papaioannou

Transaction Report:

Dear Qingfeng,

Thank you for submitting your manuscript EMBOJ-2024-116574 for consideration by The EMBO Journal. It has now been seen by three experts in the field, and we have received the full set of their reports, which are included below.

As you will see, all referees have provided thorough and detailed reports, and they are broadly supportive of the work. They point out that the findings are novel and significant, and they all agree that the data are of good quality and the conclusions, especially of the structural part of the investigation, are well-supported by the presented results. However, they also identify a number of limitations, particularly with regard to the functional analyses, and they provide a number of constructive suggestions for the improvement of the study and the manuscript.

Given the referees' positive comments and recommendations I would like to invite you to submit a revised version of your manuscript, addressing the comments of all three reviewers. I should add that it is EMBO Journal policy to allow only a single round of major revision, and acceptance of your manuscript will therefore depend on the completeness of your responses in this revised version. If you have any questions or comments, we can discuss further in a video call, if you would like to.

We generally allow three months as standard revision time (25th May 2024). As a matter of policy, competing manuscripts published during this period will not negatively impact our assessment of the conceptual advance presented by your study. However, we request that you contact us as soon as possible upon publication of any related work, to discuss how to proceed. Should you foresee a problem in meeting this three-month deadline, please let us know in advance and we may be able to grant an extension.

Thank you for the opportunity to consider your work for publication in The EMBO Journal. I look forward to your revision.

Best regards,

Ioannis

Instructions for preparing your revised manuscript

1. When you are ready to submit the revision, please upload:

- A Word file of the manuscript text (including legends of main Figures, EV Figures and Tables). Please make sure that changes are highlighted (or "tracked") to be clearly visible.

- Individual production-quality figure files (one file per figure). When assembling your figures, please refer to our figure preparation guidelines in order to ensure proper formatting and readability in print as well as on screen:

If the data shown in a figure are obtained from n {less than or equal to} 2, please use scatter plots showing the individual data points.

- i. the name of the statistical test used to generate error bars and P values
- ii. the number (n) of independent experiments (please specify technical or biological replicates) underlying each data point (discussion of statistical methodology can be reported in the Materials and Methods section, but figure legends should contain a basic description of n , P , and the test applied)
- iii. the nature of the bars and error bars (s.d., s.e.m.).

- A point-by-point response to the referees' comments, with a detailed description of the changes made (as a word file). All referees' concerns must be fully addressed and their suggestions taken on board. When preparing your letter of response to the referees' comments, please bear in mind that this will form part of the Review Process File and will therefore be available online to the community. Please note that you have the possibility to opt out of the transparent process at any stage prior to publication

by letting the editorial office know (contact@embojournal.org); if you do opt out, the Review Process File link will point to the following statement: "No Review Process File is available with this article, as the authors have chosen not to make the review process public in this case.". For more details on our Transparent Editorial Process, please visit our website: <https://www.embopress.org/page/journal/14602075/authorguide#transparentprocess>

- Expanded View (EV) files (replacing Supplementary Information) that are collapsible/expandable online. A maximum of 5 EV Figures can be typeset. EV Figures should be cited as "Figure EV1, Figure EV2" etc. in the text, and their respective legends should be included in the manuscript file after the legends of regular figures. See detailed instructions regarding Expanded View files here:

- For the figures that you do NOT wish to display as Expanded View figures, they should be bundled together with their legends in a single PDF file called "Appendix", which should start with a short Table of Contents (including page numbers). Appendix figures should be referred to in the main text as: "Appendix Figure S1, Appendix Figure S2" etc. Please see detailed instructions here: <https://www.embopress.org/page/journal/14602075/authorguide#expandedview>

- A complete author checklist, which you can download from our author guidelines (<https://www.embopress.org/page/journal/14602075/authorguide>). Please note that the checklist will also be part of the Review Process File.

2. Please note that no statistics should be calculated and shown in Figures if $n=2$.

3. Before submitting your revision, primary datasets (and computer code, where appropriate) produced in this study need to be deposited in appropriate public databases (see <https://www.embopress.org/page/journal/14602075/authorguide#dataavailability>). In particular, we would kindly ask you to deposit all new structural data.

Please remember to provide in the Data availability section of your revised manuscript reviewer passwords if the datasets are not yet public.

The accession numbers and database should be listed in a formal "Data availability" section (placed after Materials and Methods) that follows the model below (see also <https://www.embopress.org/page/journal/14602075/authorguide#dataavailability>):

Data availability

- RNA-seq data: Gene Expression Omnibus GSE46843 (<https://www.ncbi.nlm.nih.gov/geo/query/acc.cgi?acc=GSE46843>)
- [data type]: [name of the resource] [accession number/identifier/doi] ([URL or identifiers.org/DATABASE:ACCESSION])

*** Note: all links should resolve to a page where the data can be accessed. ***

*** Note: the Data Availability Section is restricted to new primary data that are part of this study. In case you have no data that require deposition in a public database, please state so instead of referring to the database: "Our study includes no data deposited in public repositories." under the heading "Data availability". ***

4. Please check that the title and the abstract of the manuscript are brief, yet explicit, even to non-specialists. The length of the title should not exceed 100 characters, and the abstract should be a single paragraph not exceeding 175 words.

5. Please also note our reference format: <https://www.embopress.org/page/journal/14602075/authorguide#referencesformat>.

7. Please remember: digital image enhancement is acceptable practice, as long as it accurately represents the original data and conforms to community standards. If a figure has been subjected to significant electronic manipulation, this must be noted in the figure legend or in the "Materials and Methods" section. The editors reserve the right to request original versions of figures and the original images that were used to assemble the figure.

8. Our journal encourages inclusion of data citations in the reference list to directly cite datasets that were obtained from public databases. Data citations in the article text are distinct from normal bibliographical citations and should directly link to the database records from which the data can be accessed. In the main text, data citations are formatted as follows: "Data ref: Smith et al, 2001" or "Data ref: NCBI Sequence Read Archive PRJNA342805, 2017". In the Reference list, data citations must

be labeled with "[DATASET]". A data reference must provide the database name, accession number/identifiers, and a resolvable link to the landing page from which the data can be accessed at the end of the reference. Further instructions are available at: <https://www.embopress.org/page/journal/14602075/authorguide#referencesformat>.

9. We request authors to consider both actual and perceived competing interests. Please review our policy (<https://www.embopress.org/page/journal/14602075/authorguide#conflictsofinterest>) and update your competing interests statement if necessary. Please name this section 'Disclosure and competing interests statement' and place it after the Acknowledgements section.

10. Please note that all corresponding authors are required to provide an ORCID ID upon submission of a revised manuscript (<https://orcid.org/>). Please find instructions on how to link your ORCID ID to your account in our manuscript tracking system in our Author guidelines (<https://www.embopress.org/page/journal/14602075/authorguide#authorshipguidelines>).

11. We use CRediT to specify the contributions of each author in the journal submission system. CRediT replaces the author contribution section, which should be removed from the manuscript. Please use the free text box to provide more detailed descriptions. See also guide to authors: <https://www.embopress.org/page/journal/14602075/authorguide#authorshipguidelines>.

13. We would also welcome the submission of cover suggestions or motifs to be used by our Graphics Illustrator in designing a cover.

14. Please use the link below to submit your revision:
<https://emboj.msubmit.net/cgi-bin/main.plex>

Referee #1:

This article describes 3 cryo-EM structures of the homomeric ACR-23 pentameric receptor from *c. elegans*. So far, several structures of mammalian nAChRs have been published, but none from invertebrate. ACR-23 is homologous, while distant, to the $\alpha 7$ receptor from animals. The 3 structures are important in several aspects, in particular ACR-23 and other worm nAChRs are the target of anti-helminthic drugs whose precise mechanism of action remains elusive. In addition, the model nematode *c. elegans* has been extensively used as a model to explore nAChR cellular distribution and signaling (see Bessereau work). The paper presents structures in apo (likely a resting state), and in complex with two anti-helminthic drugs betaine (likely a pre-active or desensitized state), and betaine+monepantel (likely an active state with an open channel). While the structures show the classical 3D fold of nAChRs, they provide new information related to this worm receptor:

1/ most importantly, the structures resolve the binding site and pose of the drugs within the structure, betaine occupying the orthosteric site, and monepantel a site at the interface between TMD helices that overlaps with the site of ivermectin on GlyRs and GluClR. The quality of the structures and the density maps appear good, allowing investigation of the detailed molecular contacts mediating binding. Key residues have been mutated suggesting impaired binding in TEVC experiments. In addition, mutations of residues contributing to monepantel binding have been investigated in behavioral tests on the worm, directly demonstrating that the identified binding site mediates monepantel effect in vivo.

2/ ACR-23 structures show specific structural features at the level of the intracellular domain and at the top of M4, with local architectures different from mammalian nAChR. These may be relevant in the future to understand the contribution of these poorly understood regions to signal transduction.

Overall, the paper is clearly presented, the data, especially cryo-EM, are of good quality, and the findings of the paper are novel in several aspects, with a clear interest in drug-design of new drugs.

Comments:

1/ The analysis shown in figure 4e and 5d indicates that mutation of key residues involved in betaine and monepantel binding site markedly decrease the currents these drugs elicit. While coherent with the cryo-EM data, these experiments are not conclusive since the mutations by themselves could impair the expression of the receptors at the cell surface to decrease the currents, without impacting ligand binding. These experiments thus lack an important control for receptor expression. I suggest that all mutants should be tested for the response to both betaine and monepantel, monepantel controlling the functional expression of betaine mutants, and conservely betaine controlling the functional expression of monepantel mutants.

2/ the functional annotation of the structures is a recurrent issue in nAChR structural biology. While authors show a superimposition of the apo structures of ACR-23 and alpha7 in supp fig 7, it would be valuable to see more superimpositions between the ACR-23 and the various alpha7 conformations solved in nanodisc and detergent. This would document the similarities and differences in gating mechanisms and further help annotate the ACR-23 structures to physiological states.

Referee #2:

This is an important study of the helminth ACR-23 ligand-gated ion channel detailing the structural basis of channel activation by the orthosteric agonist betaine and allosteric activator, monepantel. While the ligand binding sites and conformational changes associated with activation are similar to other channels, the structures are of very high resolution providing confident identification of the ligands and their binding sites. This study is significant because it yields the first high resolution structures of a ligand-gated ion channel, ACR-23, in the nematode-specific DEG-3 subfamily. In particular, the high-resolution information of the monepantel binding site in ACR-23 explains the structure-activity relationships of other AAD antihelminth drugs. Since ACR-23 is frequently mutated to produce resistance to AADs, such structural information may be useful to identify new agonists/allosteric modulators that can be used as antihelmintics.

The study is well executed, and the conclusions are generally supported by the data especially the structures. A major concern of the present submission is that there is no indication of the deposition and validation of the structures through the PDB system. Based on the refinement statistics in Table S1, it appears the authors have generated suitable models from the structures which have excellent resolution. However, validation reports as well as the maps and models should be provided for review.

In addition, I have some recommendations for the analysis and interpretation of the functional data to strengthen the conclusions of the paper regarding the structure-function relationships.

1) In Figure 1, the dose response curves should ideally be normalized to maximum response to get the best assessment of EC50, as the peak current from each cell also depends on the number of channels in the plasma membrane. Similarly, in Fig. 1D, are the data with and without betaine data from the same cell/recording? Ideally, the data should be plotted as normalized to maximum response in each patch relative to the betaine+monepantel condition in order to appropriately convey the enhanced effect of both drugs on activation. Similarly, Supplementary Fig. 2 is insufficient to show that the EM construct did not produce a shift in the potency of betaine and monepantel activation. This may be relevant if the EM construct produces a right shift in sensitivity to betaine such that 1.5 mM is insufficient to fully activate the channel. This may lead one to conclude that the betaine-bound structure is a pre-active intermediate instead of a desensitized conformation.

2) Channel activation by betaine is quite slow based on Figure 1. In the representative currents in Fig. 1a for 100 micromolar or 1000 micromolar betaine, it seems that agonist activation is not long enough to assess peak response; the current is still rising upon removal of agonist. This may lead to a right shift in the EC50. Related to this, channel activation to betaine is much slower than what was reported in the 2013 Nat Neuroscience paper. In that study, 1 mM betaine produced activation in <1s followed by significant desensitization over 20s. Furthermore, the EC50 in that study was 1 mM in contrast to 34 micromolar in the current study. This may be a consequence of differences in the expression systems used (oocyte versus HEKs), and the authors should consider commenting on this when stating that the e-phys recordings recapitulate previous TEVC results. Alternatively, is it possible that the slow activation observed in this study is due to slow solution exchange. Have the authors tested the exchange time for the perfusion system?

3) As noted above, there is no indication from the recordings shown in this paper that betaine-evoked currents desensitize and there is no measure of the extent of desensitization. If the authors conclude that the betaine-bound structure represents a desensitized conformation (page 7 line 185), then what is the experimental evidence for this? Similarly, if the authors conclude that the betaine+monepantel structure is an open conformation, is this consistent with functional data? The extent of desensitization in recordings of ACR-23 responses to betaine+monepantel are not demonstrated in this study. Based on the results from the 2013 Nat Neuroscience paper, it appears that in oocytes, betaine produces significant desensitization while monepantel+betaine produces larger steady-state currents so the authors may want to comment on this. Overall, there needs to be better characterization of the functional state of ACR-23 in the presence of agonists at steady-state to annotate the structures. One may also consider that structures of the channel in DDM and CHS may not recapitulate the lipid membrane environment in a cell altering the conformations observed in this structural analysis.

4) For analysis of the mutants in the betaine binding site (Fig. 4e), it is possible that the reduced current is simply due to decreased surface expression. The authors may consider this by assaying surface expression, or testing whether the mutations decrease agonist potency (for those mutants in which there is detectable current). Alternatively, the authors could examine peak responses to high concentration of betaine versus monepantel since responses to betaine in the mutants may be selectively attenuated compared to responses to monepantel. This same point also applies to the monepantel binding site mutants.

The following are some minor concerns for the authors to consider:

1) In Figure 1, the number of replicates for each experiment should be indicated, if they are all not triplicates.

2) In page 5 line 132, please change "High quality electron density maps" to "High quality cryo-EM density maps". Same is true for the legend of Supplementary Fig. 6, page 7 line 197, and page 9 line 251.

- 3) For page 10 lines 294-295, the *in vivo* data does not specifically argue for the importance of the mutated residues in coordinating monepantel. This is because the authors have not ruled out the possibility that the mutations prevent channel gating or decrease channel expression.
- 4) The text indicates that Supplementary Fig. 6 shows models of glycans but it is not clear from the figure if any glycans are being shown. Consider referencing Fig. 2 here.
- 5) Can the authors indicate either in Fig. 3 or in the text which of the residues form the narrowest constriction in each of the structures?
- 6) It appears that Fig. 4i and 4j are mixed up in terms of the description in the legend and what is shown. Also, it may be helpful if the loops are labeled in Fig. 4i. In Fig. 4f, it is not clear what the small box around M4 is indicating.
- 7) Regarding page 9 line 242, I believe that proline-aromatic interactions are not considered pi-pi, but rather a CH/pi interaction between the pi aromatic residue and the polarizable CH bonds in proline.
- 8) What was the concentration of the monepantel stock in DMSO which was used to make a final concentration of 5 mM? Specifically, what is the final concentration of DMSO in the sample? Please indicate this in the methods.

Referee #3:

The manuscript by Fenglian Liu and co-authors reports on the structural study of ACR-23, a nematode pentameric ligand-gated ion channel. Three structures were obtained, including a complex with monepantel, a compound used to control pathogenic helminths. The structures are complemented with functional experiments that define some of the channel properties and explore the local determinants of the effect of compounds in the orthosteric and the allosteric sites. In addition, *in vivo* experiments confirm that ACR-23 is one of the main targets of monepantel.

I am a structural biologist specialised in pentameric ligand-gated ion channels, and I found that the study was well executed on the structural part and brought a lot of novelty. I recommend the manuscript to be published. I have a couple of major remarks that should be dealt with, and an ensemble of less important ones.

Major remark #1. Desensitisation, betaine-bound structure conformation.

Line 185: "Therefore, given the fast decay of betaine-evoked currents, our betaine-bound ACR-23 structure probably represents the desensitized state". The betaine-bound structure is here assigned to the desensitized states. Yet in Figure 1 or Supplementary Figure 2, no desensitisation seems to be taking place in electrophysiology experiments. If the bars for betaine application are correct, the current are progressing throughout the application and only start to decay when the applications end. This is de-activation, not desensitisation. The absence of apparent desensitisation is by itself an interesting feature, which I thought should be discussed in the section describing the functional results.

At that point I went to see references [10] and [12]. In [12], betaine was causing desensitisation in TEVC, as was choline in [10] (but choline is described as not being an agonist in [12]). I also noted the difference between the betaine EC50 reported in the manuscript (34 μ M) and the one from [12], 1.4 mM. While it is not uncommon to see different EC50 in oocytes versus HEK cells, here we're talking of a 40-fold difference.

However, the text merely mentions that electrophysiology recordings "showed general agreement with previous results" (line 94, and a similar sentence line 320). After my quick browsing through those previous results, that sentences sounded misleading and should be corrected. The discrepancies regarding desensitisation and EC50 have to be underlined, and explained if possible.

Coming back to the betaine-bound structure, the assignment of structures to physiological states has been a central concern of a large fraction of the pLGIC structural biology papers. The location of the desensitisation gate in cationic members is also not fully established (probably at the 9' level). Here with the functional data showing no desensitisation in HEK cells, if the structure is assigned to a state (is it necessary?), it should be with a very cautious wording.

Major remark #2. Permeation

Less important than #1, as this is merely a presentation/analysis issue, but still. In the paragraph describing the pore profiles (starting line 171) a lot of emphasis is put on the static radius, which is used to define if ions can go through (several mentions of the constriction radius). Maybe one of the lessons learned from the accumulation of pLGICs structure is the limited relevance of static radius to define if the pore is open or closed. First we're here speaking about a heteropentamer refined in C5, when each constriction point is thus defined by 5 equivalent rotamers of side chains, a situation probably not too representative of the pore dynamics (see for instance Cymes and Grosman *Nat Chem Biol* 2012). Second the activation gate has been convincingly described as working by a wetting/dewetting mechanism (molecular dynamics), and it is probably more relevant to consider radius + orientation of 9' and other hydrophobic residues in/out the pore + comparison of M2 conformations to other structures when dealing with permeation.

Statements such as "a constriction of this radius is too narrow for that ion" makes me skeptical/wary every time.

This remark echoes Minor #6: the analysis of permeation would benefit of added comparisons. E.g. the pore in the structure X is quite similar to published structure Y, which has been clearly characterised as open/closed. I'm not sure which ones are the best pLGICs to compare to, but it's probably worth expanding beyond the comparison with alpha7 nAChR.

Intermediate remark #3: More details on methods

The co-infection with RIC-3 should be described in details. What was the ratio of RIC3:ACR-23 viruses used? Was RIC-3 also in a pFastBac vector, ie under the control of the same promoter as ACR-23? If this was an important point to successfully express the protein, it should be documented in a SI figure, with comparisons (WB, FSEC, ...) between ACR-23 alone and co-expressed with RIC-3.

Minor remarks.

#4

Is the Y unit of Fig. 1b ok? I see a dose/response with error bars, so I guess this was obtained from several cells (the caption mentions $n > 3$). It is then surprising that the Y axis is labeled in pA and not on a normalised scale from 0 to 1, as it implies that the different cells had the same maximum current upon application of a saturating concentration of betaine. I don't know much of electrophysiology in HEK cells, so it may be standard not to have to normalise as one would do in TEVC.

Same remark for Fig. 1D

#5

Figure 4. C224 is involved in direct interaction with the ligand. A CC motif in the loop C is typical of alpha type subunits of nAChR; there does not seem to be a consistent pattern whether those two cysteines are linked in existing structures (I'm not sure). How was it for ACR-23? Unambiguous densities?

Looking at the structures, there is a second water molecule not depicted in Fig. 4. Is it because it is irrelevant for betaine binding?

Comparisons with other agonist-bound pLGIC are included in SI9 and briefly described around line 209. Those comparisons could be more extensive, both in terms of text and figures, with close-ups with superimpositions.

#6

In general, comparison of ACR-23 with other pLGIC was mostly limited to the alpha7 nAChR. As an interesting point of the manuscript, and of similar structural work, is always what in the gating transition is specific to any member (here ACR-23) versus what is shared by the whole family of pLGIC, it could be beneficial to make multiple comparisons with several pLGICs. I'm agnostic as to what needs to be shown in a revised version, maybe nothing more, but if other comparisons were not done, I suggest trying with homomers that have several known states, e.g. GlyR, 5HT3R, GABA rho1, bacterial channels, ...

#7

beautiful densities for monepantel, comparison of the site with other pLGIC is well done (line ~260 and after). Local sequence alignments of M1(-) and M2-M3(+) could help in SI12. As the authors already included comparison with a7, it would be a plus to include extra superimpositions with the new a7 structures with PAM that were published during the review process.

<https://doi.org/10.1016/j.cell.2024.01.032>

#8

Fig. 5D. The effect of mutations is in line with the ligand pose, any specific idea about the three mutants that preserve function? A310F in particular looks interesting as it introduces a bulky side chain that could close the cavity once monepantel is there?

#9

Around line 291, the authors explains their in vivo findings, where only 1/3rd of monepantel activity is maintained upon knocking-out ACR-23. This is cool, and could be easily pushed a bit further by discussing what the remaining targets could be. Based on their discovery of the binding site and a multiple sequence alignment, can the authors predict/hypothesize which other pLGIC have a quite similar intersubunit pocket? I guess this also related to ref [2], which I did not have time to read

#10

Fig 5f. Maybe a new SI panel, showing the same type of superimposition but for betaine and betaine-monepantel would be helpful. In particular, I wondered if the monepantel site was pre-existing in the betaine bound state. This is relevant to discuss the allosteric (or not) mechanism of ACR-23.

#11

lines 343-351. Why such an emphasis on the latch? It's a distinct secondary structure element in alpha7 and ACR-23, yes, but one could also point out that some subunits of the Torpedo muscle-type nAChR have distinct C-ters... I am not sure how important those differences are at that level

#12

The construct design is not included in the Methods and should be added. The reader can guess some info on the sequence alignment of Figure S11 but it would be better if explicit. Also interesting for people in the pLGIC field (but generally not detailed in many papers) is the design strategy that was followed. How did the authors choose where to insert BRIL? Did they consider or test other fusions (GFP is quite commonly inserted in the ICD as well)

#13

line 592, mention the final DMSO concentration after adding monepantel

#14

The cryoEM data treatment section is fine, but out of curiosity, I wonder why the final sets of particles are so big. For instance ~500k particles are selected for 3D and out of them ~400k make it to the final set. This is very ok, yet I've often seen attrition rate much bigger in cryoEM papers of pLGICs, with people trashing 80% of their particles during 3D classifications. Does that mean that here the sample is very homogeneous (the local res pictures shown in SI tend to indicate that), maybe more than some other pLGICs? Have the authors tried to heavily classify and see how the reconstruction was with the best ~50k particles? Given the large set of particles they have, have they tried to refine in C1, or perform symmetry-expansion + classifications to reveal potential asymmetries, or look at the flexibility?

#15

At the end of manuscript, I realised that the helix MB in the ICD was not described. Does it deserve a mention? also because the Mx-MB connection is at the subunit interface and might be moving during gating.

Typos and very minor things

- A repetition between abstract and intro (1st sentence!)
- I don't know much about anti-helminthic drugs, and it will be the same for other readers. Maybe the authors could give a couple extra info (size of the market, human vs livestock vs crop protection)
- line 84, maybe avoid the use of atomic resolution (given past heated debates in cryoEM mailing lists). It is sufficient to tell the readers that ligands pose are unambiguous
- line 85, typo, "sited-directed"
- In the workflow figures, some reconstructions are flipped (e.g Supplementary Fig. 4C). While this is not big deal, you may consider to put all of them with the correct hand

Inspection of the structures and maps

I briefly inspected the models and maps at the global level in ChimeraX and looking at ligands in Coot and everything seemed satisfying. Congratulations on the beautiful structures! Some lipid densities are good and could have been described in the paper (e.g. the one lining monepantel outside of its cavity, but not only)

Referee #1:

This article describes 3 cryo-EM structures of the homomeric ACR-23 pentameric receptor from *c. elegans*. So far, several structures of mammalian nAChRs have been published, but none from invertebrate. ACR-23 is homologous, while distant, to the alpha7 receptor from animals. The 3 structures are important in several aspects, in particular ACR-23 and other worm nAChRs are the target of anti-helminthic drugs whose precise mechanism of action remains elusive. In addition, the model nematode *c. elegans* has been extensively used as a model to explore nAChR cellular distribution and signaling (see Bessereau work).

The paper presents structures in apo (likely a resting state), and in complex with two anti-helminthic drugs betaine (likely a pre-active or desensitized state), and betaine+monepantel (likely an active state with an open channel). While the structures show the classical 3D fold of nAChRs, they provide new information related to this worm receptor:

1/ most importantly, the structures resolve the binding site and pose of the drugs within the structure, betaine occupying the orthosteric site, and monepantel a site at the interface between TMD helices that overlaps with the site of ivermectin on GlyRs and GluClR. The quality of the structures and the density maps appear good, allowing investigation of the detailed molecular contacts mediating binding. Key residues have been mutated suggesting impaired binding in TEVC experiments. In addition, mutations of residues contributing to monepantel binding have been investigated in behavioral tests on the worm, directly demonstrating that the identified binding site mediates monepantel effect in vivo.

2/ ACR-23 structures show specific structural features at the level of the intracellular domain and at the top of M4, with local architectures different from mammalian nAChR. These may be relevant in the future to understand the contribution of these poorly understood regions to signal transduction.

Overall, the paper is clearly presented, the data, especially cryo-EM, are of good quality, and the findings of the paper are novel in several aspects, with a clear interest in drug-design of new drugs.

Reply: We appreciate the reviewer's positive comments and constructive suggestions. We have collectively addressed all the reviewer's concerns by performing suggested experiments and reanalyzing data.

Comments:

1/ The analysis shown in figure 4e and 5d indicates that mutation of key residues involved in betaine and monepantel binding site markedly decrease the currents these drugs elicit. While coherent with the cryo-EM data, these experiments are not conclusive since the mutations by themselves could impair the expression of the receptors at the cell surface to decrease the currents, without impacting ligand binding.

These experiments thus lack an important control for receptor expression. I suggest that all mutants should be tested for the response to both betaine and monepantel, monepantel controlling the functional expression of betaine mutants, and conservely betaine controlling the functional expression of monepantel mutants.

Reply:

We thank the reviewer for suggestions. We have performed the experiments suggested by the reviewer.

- (1) We have checked the expression of the WT and mutant receptors using both confocal imaging and WB, and found that like the WT ACR-23, most of the mutants we generated for electrophysiological recording are expressed on plasma membrane (Appendix Fig. S7 and S10).
- (2) We have tested the response to both betaine and monepantel on all mutants, and found that while most of the betaine binding site mutants could be activated by monepantel, major portion of the monepantel binding site mutants could not be activated by betaine, albeit with similar expression level and expression pattern. We reasoned that some of the residues involved in monepantel binding are located in M2 or M2-M3 loop, which might affect activation of ACR-23 by betaine by perturbing pore opening and/or propagation of conformational changes toward the pore (In pdf file: lines 317-321).

2/ the functional annotation of the structures is a recurrent issue in nAChR structural biology. While authors show a superimposition of the apo structures of ACR-23 and alpha7 in supp fig 7, it would be valuable to see more superimpositions between the ACR-23 and the various alpha7 conformations solved in nanodisc and detergent. This would document the similarities and differences in gating mechanisms and further help annotate the ACR-23 structures to physiological states.

Following the reviewer's suggestions. we have made more comparison with various alpha7 conformations, particularly in the pore, to help annotate the ACR-23 structures to functional states (Fig. EV2). We concluded that the functional state of the betaine-bound ACR-23 structure is not clear and as suggested by other reviewers, we decided not to assign the functional state of the betaine-bound ACR-23 structure. The other two structure (apo and betaine&monepantel-bound) are confirmed to be in resting closed and open state respectively by MD simulations (Fig. 2D).

Referee #2:

This is an important study of the helminth ACR-23 ligand-gated ion channel detailing the structural basis of channel activation by the orthosteric agonist betaine and allosteric activator, monepantel. While the ligand binding sites and conformational changes associated with activation are similar to other channels, the structures are of very high resolution providing confident identification of the ligands and their binding sites. This study is significant because it yields the first high resolutions structures of a ligand-gated ion channel, ACR-23, in the nematode-specific DEG-3 subfamily. In

particular, the high-resolution information of the monepantel binding site in ACR-23 explains the structure-activity relationships of other AAD antihelminth drugs. Since ACR-23 is frequently mutated to produce resistance to AADs, such structural information may be useful to identify new agonists/allosteric modulators that can be used as antihelmintics.

The study is well executed, and the conclusions are generally supported by the data especially the structures. A major concern of the present submission is that there is no indication of the deposition and validation of the structures through the PDB system. Based on the refinement statistics in Table S1, it appears the authors have generated suitable models from the structures which have excellent resolution. However, validation reports as well as the maps and models should be provided for review.

In addition, I have some recommendations for the analysis and interpretation of the functional data to strengthen the conclusions of the paper regarding the structure-function relationships.

We appreciate the reviewer's positive comments and constructive suggestions. We have collectively addressed all the reviewer's concerns by performing suggested experiments and reanalyzing data. Models and maps have been deposited and validation reports are now provided.

1) In Figure 1, the dose response curves should ideally be normalized to maximum response to get the best assessment of EC50, as the peak current from each cell also depends on the number of channels in the plasma membrane. Similarly, in Fig. 1D, are the data with and without betaine data from the same cell/recording? Ideally, the data should be plotted as normalized to maximum response in each patch relative to the betaine+monepantel condition in order to appropriately convey the enhanced effect of both drugs on activation. Similarly, Supplementary Fig. 2 is insufficient to show that the EM construct did not produce a shift in the potency of betaine and monepantel activation. This may be relevant if the EM construct produces a right shift in sensitivity to betaine such that 1.5 mM is insufficient to fully activate the channel. This may lead one to conclude that the betaine-bound structure is a pre-active intermediate instead of a desensitized conformation.

Reply:

We appreciate the reviewer's critical comment. We have normalized the dose response curves in Fig. 1 to maximum response.

For Appendix Fig. S2, we have conducted extra experiments and determined the EC50 of betaine and monepantel, which confirmed that the EM construct and WT are similar in EC50 for both compounds. We agree that the functional state of betaine-bound structure is not clear and does not seem to be desensitized state, as the desensitization gate (-I' or intracellular gate that may close during desensitization according to the dual gate model (PMC5978336)) in this structure is not constricted. Therefore, we decided not to assign the functional state of the betaine-bound ACR-23

structure for now.

2) Channel activation by betaine is quite slow based on Figure 1. In the representative currents in Fig. 1a for 100 micromolar or 1000 micromolar betaine, it seems that agonist activation is not long enough to assess peak response; the current is still rising upon removal of agonist. This may lead to a right shift in the EC₅₀. Related to this, channel activation to betaine is much slower than what was reported in the 2013 Nat Neuroscience paper. In that study, 1 mM betaine produced activation in <1s followed by significant desensitization over 20s. Furthermore, the EC₅₀ in that study was 1 mM in contrast to 34 micromolar in the current study. This may be a consequence of differences in the expression systems used (oocyte versus HEKs), and the authors should consider commenting on this when stating that the e-phys recordings recapitulate previous TEVC results. Alternatively, is it possible that the slow activation observed in this study is due to slow solution exchange. Have the authors tested the exchange time for the perfusion system?

Following the reviewer's comments and constructive suggestions, we have repeated the experiments in Fig. 1, during which betaine was applied for longer time until the activation to reach the peak. Now the measured EC₅₀ is 34.43 μM, which is still much lower than previous reported. We have now changed wording on the part describing e-phys recordings and attributed the differences in activation speed and EC₅₀ to the expression systems used. We used a Rapid solution changer (RSC 200, BioLogic), which could reach 2ms of extremely fast solution change by switching 8-10 solution tubes (In pdf file: line 729-730).

3) As noted above, there is no indication from the recordings shown in this paper that betaine-evoked currents desensitize and there is no measure of the extent of desensitization. If the authors conclude that the betaine-bound structure represents a desensitized conformation (page 7 line 185), then what is the experimental evidence for this? Similarly, if the authors conclude that the betaine+monepantel structure is an open conformation, is this consistent with functional data? The extent of desensitization in recordings of ACR-23 responses to betaine+monepantel are not demonstrated in this study. Based on the results from the 2013 Nat Neuroscience paper, it appears that in oocytes, betaine produces significant desensitization while monepantel+betaine produces larger steady-state currents so the authors may want to comment on this. Overall, there needs to be better characterization of the functional state of ACR-23 in the presence of agonists at steady-state to annotate the structures. One may also consider that structures of the channel in DDM and CHS may not recapitulate the lipid membrane environment in a cell altering the conformations observed in this structural analysis.

Reply:

We appreciate the reviewer's constructive suggestions. For betaine-bound structures, we have performed new experiments, which show that there is indeed desensitization for betaine-activated

currents, but much slower when compared with that reported in 2014 Nat Neuroscience paper. Given the ambiguity, we decided not to assign its functional state for now. Instead, “betaine-bound state” is used.

Our results shown that in our system, application of monepantel+betaine also produces larger steady-state currents (Fig. 1E). To better characterize the functional states of ACR-23 structures, we performed MD simulation and analyzed the wetting and ion conducting events of ACR-23 pores of our 3 structures, which support our claim that monepantel+betaine-bound structure is in open state and the other two structures are in non-conductive states (Fig. 3D).

It is a known concern that structures determined in detergent may not recapitulated the lipid environment, which cannot be ruled out in our study as well.

4) For analysis of the mutants in the betaine binding site (Fig. 4e), it is possible that the reduced current is simply due to decreased surface expression. The authors may consider this by assaying surface expression, or testing whether the mutations decrease agonist potency (for those mutants in which there is detectable current). Alternatively, the authors could examine peak responses to high concentration of betaine versus monepantel since responses to betaine in the mutants may be selectively attenuated compared to responses to monepantel. This same point also applies to the monepantel binding site mutants.

Reply:

We thank the reviewer for suggestions. This point has also been brought up by other reviewers.

We have checked the expression of the WT and mutant receptors using both confocal imaging and WB, and found that like the WT ACR-23, most of the mutants we generated for electrophysiological recording have similar expression level and pattern as WT. We have presented these data (Appendix Fig. S7, S10) and added some discussions about this (In pdf file: lines 256-257; line 317).

We have tested the response to both betaine and monepantel on all mutants, and found that all betaine site mutants except Y116A could be activated by monepantel, whereas for monepantel site mutants, only 4 out of 11 could be activated by betaine. We reasoned that, since residues forming the monepantel site are all located in TMD, and some are on the pathway through which the betaine induced conformational changes are propagated towards the ion-conduction pore in TMD, these mutations impaired channel activation by betaine (In pdf file: lines 319-321).

The following are some minor concerns for the authors to consider:

1) In Figure 1, the number of replicates for each experiment should be indicated, if they are all not triplicates.

Reply: The number of replicates is now indicated in figure legend.

2) In page 5 line 132, please change "High quality electron density maps" to "High quality cryo-EM density maps". Same is true for the legend of Supplementary Fig. 6,

page 7 line 197, and page 9 line 251.

Reply: Changes have been made accordingly (In pdf file: line 151; legend of Appendix Fig. S6; line 229; line 294).

3) For page 10 lines 294-295, the in vivo data does not specifically argue for the importance of the mutated residues in coordinating monepantel. This is because the authors have not ruled out the possibility that the mutations prevent channel gating or decrease channel expression.

Reply: We thank the reviewer for critical comments. We have now performed WB on total lysate of WT and mutant worms, which showed that these mutations did not significantly affect channel expression when compared with WT (In pdf file: line 344. Appendix Fig. S12D).

4) The text indicates that Supplementary Fig. 6 shows models of glycans but it is not clear from the figure if any glycans are being shown. Consider referencing Fig. 2 here.

Reply: The densities for glycans have now been shown in Supplementary Fig. 6 (now Appendix Fig S6).

5) Can the authors indicate either in Fig. 3 or in the text which of the residues form the narrowest constriction in each of the structures?

Reply: The residues forming the narrowest constriction are now indicated in Fig. 3B.

6) It appears that Fig. 4i and 4j are mixed up in terms of the description in the legend and what is shown. Also, it may be helpful if the loops are labeled in Fig. 4i. In Fig. 4f, it is not clear what the small box around M4 is indicating.

Reply: We thank the reviewer for suggestions. After double check, descriptions for Fig. 4i and 4j were not mixed up. We have now labeled the loops in Fig. 4I. For clarity, we labeled the small boxes (with a letter above the box) in Fig. 4F to indicate the regions shown in Fig. 4G, I and J.

7) Regarding page 9 line 242, I believe that proline-aromatic interactions are not considered pi-pi, but rather a CH/pi interaction between the pi aromatic residue and the polarizable CH bonds in proline.

Reply: We have now used the correct term in the text (In pdf file: line 282).

8) What was the concentration of the monepantel stock in DMSO which was used to make a final concentration of 5 mM? Specifically, what is the final concentration of DMSO in the sample? Please indicate this in the methods.

Reply: We have now indicated in the methods that the final concentration of DMSO in the sample is 10% (v:v) (In pdf file: line 668).

Referee #3:

The manuscript by Fenglian Liu and co-authors reports on the structural study of ACR-23, a nematode pentameric ligand-gated ion channel. Three structures were obtained, including a complex with monepantel, a compound used to control pathogenic helminths. The structures are complemented with functional experiments that define some of the channel properties and explore the local determinants of the effect of compounds in the orthosteric and the allosteric sites. In addition, in vivo experiments confirm that ACR-23 is one of the main targets of monepantel.

I am a structural biologist specialised in pentameric ligand-gated ion channels, and I found that the study was well executed on the structural part and brought a lot of novelty. I recommend the manuscript to be published. I have a couple of major remarks that should be dealt with, and an ensemble of less important ones.

We thank the reviewer's positive comments and constructive suggestions. We have collectively addressed all the reviewer's concerns by performing suggested experiments and reanalyzing data.

The following are our point-to-point responses to the reviewer's comments

Major remark #1. Desensitisation, betaine-bound structure conformation.

Line 185: "Therefore, given the fast decay of betaine-evoked currents, our betaine-bound ACR-23 structure probably represents the desensitized state". The betaine-bound structure is here assigned to the desensitized states. Yet in Figure 1 or Supplementary Figure 2, no desensitisation seems to be taking place in electrophysiology experiments. If the bars for betaine application are correct, the current are progressing throughout the application and only start to decay when the applications end. This is de-activation, not desensitisation. The absence of apparent desensitisation is by itself an interesting feature, which I thought should be discussed in the section describing the functional results.

Reply:

We thank the reviewer for critical comments. We have repeated these measurements with longer duration of betaine application and found that there is indeed desensitization for betaine-evoked ACR-23 currents, but much slower than reported in 2013 Nat Neuroscience paper. Given the ambiguity, we followed your suggestion and leave the functional state of betaine-bound ACR-23 structure undefined (In pdf file: lines 213-214; lines 378-379).

At that point I went to see references [10] and [12]. In [12], betaine was causing desensitisation in TEVC, as was choline in [10] (but choline is described as not being

an agonist in [12]). I also noted the difference between the betaine EC50 reported in the manuscript (34 μ M) and the one from [12], 1.4 mM. While it is not uncommon to see different EC50 in oocytes versus HEK cells, here we're talking of a 40-fold difference.

Reply:

We repeated these experiments and confirmed that these differences are true. Choline could not activate ACR-23 in our system either. We do not know the exact reason underlying these differences and thought that it may be due to different systems used for recordings, which may have different post-translational modifications and contain distinct modulatory proteins (In pdf file: lines 113-115).

However, the text merely mentions that electrophysiology recordings "showed general agreement with previous results" (line 94, and a similar sentence line 320). After my quick browsing through those previous results, that sentences sounded misleading and should to be corrected. The discrepancies regarding desensitisation and EC50 have to be underlined, and explained if possible.

Reply: We have changed our wording to underline the differences and explained that these differences might be due to different systems used for recordings (In pdf file: lines 113-115 and lines 377-379).

Coming back to the betaine-bound structure, the assignment of structures to physiological states has been a central concern of a large fraction of the pLGIC structural biology papers. The location of the desensitisation gate in cationic members is also not fully established (probably at the 9' level). Here with the functional data showing no desensitisation in HEK cells, if the structure is assigned to a state (is it necessary?), it should be with a very cautious wording.

Reply: We followed your suggestion and leave the functional state of betaine-bound ACR-23 structure undefined (In pdf file: lines 213-214).

Major remark #2. Permeation

Less important than #1, as this is merely a presentation/analysis issue, but still. In the paragraph describing the pore profiles (starting line 171) a lot of emphasis is put on the static radius, which is used to define if ions can go through (several mentions of the constriction radius). Maybe one of the lessons learned from the accumulation of pLGICs structure is the limited relevance of static radius to define if the pore is open or closed. First we're here speaking about an heteropentamer refined in C5, when each constriction point is thus defined by 5 equivalent rotamers of side chains, a situation probably not too representative of the pore dynamics (see for instance Cymes and Grosman Nat Chen Biol 2012). Second the activation gate has been convincingly described as working by a wetting/dewetting mechanism (molecular dynamics), and it is probably more relevant to consider radius + orientation of 9' and other hydrophobic

residues in/out the pore + comparison of M2 conformations to other structures when dealing with permeation.

Reply:

We performed MD simulations to see wetting/dewetting and ion permeability on our ACR-23 structures. While apo and betaine-bound structure are non-conductive, betaine&nonepantel-bound structure is permeable to Na⁺ (Fig. 3D).

In addition, we performed extensive comparison of pore radius and orientation of pore-lining residues with alpha1, alpha7 and GluCl (only orientation of pore-line residues is shown for GluCl) receptors (Fig. EV2E-H), whose multiple functional states (resting/desensitized/open) are clearly defined. These comparisons support a dual gate model (PMC5978336), in which -1' (desensitization gate or intracellular gate) closes during desensitization, whereas the activation gate (9') is usually still open. In our betaine-bound ACR-23 structure, however, -1' of our betaine-bound ACR-23 structure is not the most constricted (9' is constricted instead, meaning the activation gate is still closed). Therefore, we state that our betaine-bound ACR-23 structure is unlikely in desensitized state and its functional state could not be reliably assigned for now (In pdf file: lines 2019-214).

Statements such as "a constriction of this radius is too narrow for that ion" makes me skeptical/wary every time.

Reply:

While we still keep these statements, they are followed by statements of MD simulations showing hydration and ion permeability (In pdf file: line 209, 217).

This remark echoes Minor #6: the analysis of permeation would benefit of added comparisons. E.g. the pore in the structure X is quite similar to published structure Y, which has been clearly characterised as open/closed. I'm not sure which ones are the best pLGICs to compare to, but it's probably worth expanding beyond the comparison with alpha7 nAChR.

Reply: As mentioned above, we have added extensive comparisons in Fig. EV2.

Intermediate remark #3: More details on methods

The co-infection with RIC-3 should be described in details. What was the ratio of RIC3:ACR-23 viruses used? Was RIC-3 also in a pFastBac vector, ie under the control of the same promoter as ACR-23? If this was an important point to successfully express the protein, it should be documented in a SI figure, with comparisons (WB, FSEC, ...) between ACR-23 alone and co-expressed with RIC-3.

Reply:

We thank the reviewer for suggestions. As it was reported that RIC-3 is important for folding of ACR-23, we chose to co-express RIC-3 with ACR-23. Following your suggestion, we compared the expression level and biochemical behavior of ACR-23 without and with the presence of RIC-3,

as judged by SEC (see figure below). It is now clear that RIC-3 does not seem to have any effects on successful expression and purification of ACR-23. However, since we actually co-expressed ACR-23 with RIC-3, we kept RIC-3 in the method section.

Minor remarks.

#4

Is the Y unit of Fig. 1b ok? I see a dose/response with error bars, so I guess this was obtained from several cells (the caption mentions $n > 3$). It is then surprising that the Y axis is labeled in pA and not on a normalised scale from 0 to 1, as it implies that the different cells had the same maximum current upon application of a saturating concentration of betaine. I don't know much of electrophysiology in HEK cells, so it may be standard not to have to normalise as one would do in TEVC.

Same remark for Fig. 1D

Reply: We thank the reviewer for helpful suggestions. Similar suggestions have been brought up by other reviewers. Fig. 1B&1D has now been shown on a normalized scale instead of pA.

#5

Figure 4. C224 is involved in direct interaction with the ligand. A CC motif in the loop C is typical of alpha type subunits of nAChR; there does not seem to be a consistent pattern whether those two cysteines are linked in existing structures (I'm not sure). How was it for ACR-23? Unambiguous densities?

We thank the reviewer for careful inspection. In our 3 ACR-23 structures, densities are not perfect but are OK. Two cysteines in the CC motif are not linked in ACR-23. After quickly going through a few structure biology papers about alpha type subunits of nAChR, we found that at least in alpha7 receptor and alpha subunit of a muscle-type nicotinic acetylcholine receptor, the equivalent cysteine pairs are not linked, just like in ACR-23.

Looking at the structures, there is a second water molecule not depicted in Fig. 4. Is it because it is irrelevant for betaine binding?

Reply: As the distance between the other water molecule and betaine is too far to form hydrogen-bond, it is not depicted in Fig. 4. Whether these water molecules are essential for betaine binding awaits further investigation.

Comparisons with other agonist-bound pLGIC are included in SI9 and briefly described around line 209. Those comparisons could be more extensive, both in terms of text and figures, with close-ups with superimpositions.

Reply: We have now added more comparisons with other agonist-bound pLGICs (SI9, now Fig. EV 3; In pdf file: lines 247-248).

#6

In general, comparison of ACR-23 with other pLGIC was mostly limited to the alpha7 nAChR. As an interesting point of the manuscript, and of similar structural work, is always what in the gating transition is specific to any member (here ACR-23) versus what is shared by the whole family of pLGIC, it could be beneficial to make multiple comparisons with several pLGICs.

I'm agnostic as to what needs to be shown in a revised version, maybe nothing more, but if other comparisons were not done, I suggest trying with homomers that have several known states, e.g. GlyR, 5HT3R, GABA rho1, bacterial channels, ...

Reply:

We thank the reviewer for constructive suggestions. As mentioned above, we have added extensive comparisons of pores in TMD of pLGICs in Fig. EV2. In addition, we performed comparison with other LGICs regarding the overall structure in Fig. EV1A, and regarding ECD-TMD junction in Fig. EV3F-H. We believe these additional comparisons should add some insights into the similarity and difference in the gating transition among pLGICs

#7

beautiful densities for monepantel, comparison of the site with other pLGIC is well done (line ~260 and after). Local sequence alignments of M1(-) and M2-M3(+) could

help in SI12. As the authors already included comparison with a7, it would be a plus to include extra superimpositions with the new a7 structures with PAM that were published during the review process.

<https://doi.org/10.1016/j.cell.2024.01.032>

Reply:

We thank the reviewer for positive comments and suggestions. We have added recently published alpha7 structures with PAM and local sequence alignments of M1-M3 in SI12 (now Fig. EV4B).

#8

Fig. 5D. The effect of mutations is in line with the ligand pose, any specific idea about the three mutants that preserve function? A310F in particular looks interesting as it introduces a bulky side chain that could close the cavity once monepantel is there?

Reply:

We thank the reviewer for asking. For A310F, we speculate that since entrance of the cavity for monepantel is wide enough so that A310F only partially block the entrance and monepantel still manage to enter the pocket (e.g., sliding from the intracellular leaflet). Interestingly, all these 3 mutants seem to be located at the entrance of the pocket, but not deeper in the pocket (Fig. 5B).

#9

Around line 291, the authors explains their *in vivo* findings, where only 1/3rd of monepantel activity is maintained upon knocking-out ACR-23. This is cool, and could be easily pushed a bit further by discussing what the remaining targets could be. Based on their discovery of the binding site and a multiple sequence alignment, can the authors predict/hypothesize which other pLGIC have a quite similar intersubunit pocket? I guess this also related to ref [2], which I did not have time to read

Reply:

We thank the reviewer for suggestions. It is indeed known that another nematode specific DEG-3 subfamily member ACR-20 can be activated by monepantel and is likely among the remaining targets of monepantel. We have now added statement that ACR-20 could be one the remaining targets of monepantel in the section where the *in vivo* assay results are discussed (In pdf file: line 340).

#10

Fig 5f. Maybe a new SI panel, showing the same type of superimposition but for betaine and betaine-monepantel would be helpful. In particular, I wondered if the monepantel site was pre-existing in the betaine bound state. This is relevant to discuss the allosteric (or not) mechanism of ACR-23.

Reply:

We thank the reviewer for suggestions. We have now included very extensive superimposition of

all three states (Fig. EV5). It seems that monepantel site does pre-exist partially in the betaine-bound state, but further expansion is required for formation of a full monepantel binding site. Indeed, we saw some densities that partially occupy the monepantel binding site in betaine-bound ACR-23 structure, which are absent in apo and betaine&monepantel-bound states (Fig. 1).

#11

lines 343-351. Why such an emphasis on the latch? It's a distinct secondary structure element in alpha7 and ACR-23, yes, but one could also point out that some subunits of the Torpedo muscle-type nAChR have distinct C-ters... I am not sure how important those differences are at that level

Reply:

It is known that the latch is important for alpha7. We do not know the importance of this region in ACR-23, however, given the overall structural similarity between ACR-23 and alpha7, we just want to emphasize that, unlike in alpha7, the region corresponding to the latch does not seem to play an important role in channel activation by agonist. Therefore, C-ters seem to be less conserved, and unlike Cys-loop, are not universally involved in activation.

#12

The construct design is not included in the Methods and should be added. The reader can guess some info on the sequence alignment of Figure S11 but it would be better if explicit. Also interesting for people in the pLGIC field (but generally not detailed in many papers) is the design strategy that was followed. How did the authors choose where to insert BRIL? Did they consider or test other fusions (GFP is quite commonly inserted in the ICD as well)

Reply:

We thank the reviewer for suggestions. Construct design strategy is now included in the Results section (In pdf file: line 139). We basically followed the strategy of $\alpha 7$ (In pdf file: line 138) and luckily, it worked without trying too many constructs with different fusion sites. It is less lucky for another unpublished pLGIC we are working on in our lab, which costed us a lot of efforts to find the optimal fusion site. As BRIL worked nicely, we did not try other fusions. Basically, it is still largely a try-and-error process, especially when there are no experiences from other closely related homologs.

#13

line 592, mention the final DMSO concentration after adding monepantel

Reply: The final DMSO concentration, 10% (v:v), is now mentioned in the method section (In pdf file: line 668).

#14

The cryoEM data treatment section is fine, but out of curiosity, I wonder why the final

sets of particles are so big. For instance ~500k particles are selected for 3D and out of them ~400k make it to the final set. This is very ok, yet I've often seen attrition rate much bigger in cryoEM papers of pLGICs, with people trashing 80% of their particles during 3D classifications. Does that mean that here the sample is very homogeneous (the local res pictures shown in SI tend to indicate that), maybe more than some other pLGICs? Have the authors tried to heavily classify and see how the reconstruction was with the best ~50k particles? Given the large set of particles they have, have they tried to refine in C1, or perform symmetry-expansion + classifications to reveal potential asymmetries, or look at the flexibility?

Reply:

Given the high data quality of the ACR23 sample and the ample dataset available, we were able to conduct rigorous 2D classification, resulting in the removal of numerous generic particle categories. Instead, we retained only high-quality 2D categories, allowing us to preserve approximately 80% of the final particles for subsequent 3D classification. Following your suggestions, we further performed 3D classification and reconstruction on this subset of particles, without applying symmetry. The outcome revealed that, apart from 3.2% (2.1% + 1.1%) categorized as garbage particles, the classification results for the remaining particles exhibited very similar proportions (21.1%, 24.9%, 26.7%, 24.1%, respectively). Additionally, the final reconstructed density maps demonstrated remarkable similarity, indicating the excellent homogeneity of ACR23.

At the end of manuscript, I realised that the helix MB in the ICD was not described. Does it deserve a mention? also because the Mx-MB connection is at the subunit interface and might be moving during gating.

Reply:

We thank the reviewer for suggestions. We only briefly mentioned MB in our manuscript. This region is usually disordered in other structures (usually it was MA if any part of M3-M4 loop was visible), therefore, we did not mention too much about it. According to Appendix Fig. S10, Mx-MB connection seems to undergo rigid-body movement along with most other part of TMD.

Typos and very minor things

- A repetition between abstract and intro (1st sentence!)

Reply: This sentence has been removed from abstract.

- I don't know much about anti-helminthic drugs, and it will be the same for other readers. Maybe the authors could give a couple extra info (size of the market, human vs livestock vs crop protection)

Reply: A few sentences about anthelmintics and their size of market have now been included (In pdf file: lines 47-49).

- line 84, maybe avoid the use of atomic resolution (given past heated debates in cryoEM mailing lists). It is sufficient to tell the readers that ligands pose are unambiguous

Reply: The term 'Atomic resolution' has been avoided (In pdf file: line 96).

- line 85, typo, "sited-directed"

Reply: The typo has been corrected (In pdf file: line 97).

- In the workflow figures, some reconstructions are flipped (e.g Supplementary Fig. 4C). While this is not big deal, you may consider to put all of them with the correct hand

Reply: We have corrected the hand of molecules in Supplementary Fig. 4C (now Appendix Fig. S3-5)

Inspection of the structures and maps

I briefly inspected the models and maps at the global level in ChimeraX and looking at ligands in Coot and everything seemed satisfying. Congratulations on the beautiful structures! Some lipid densities are good and could have been described in the paper

(e.g. the one lining monepantel outside of its cavity, but not only)

Reply: We thank the reviewer for careful inspection and suggestion. We have now showed some lipid-like densities in Fig. 1, and mentioned their existence in the legend and text (In pdf file: lines 154-157). Notably, some densities that partially occupy the monepantel binding site are seen in the betaine-bound ACR-23 structure, but not in apo and betaine&monepantel-bound states.

Dear Qingfeng,

Thank you again for the submission of your revised manuscript to The EMBO Journal. We have now received the comments of all three referees that were asked to re-assess your study (included below). As you will see, all referees are satisfied with the revision, acknowledge that the initially raised concerns have been successfully addressed during revision with the addition of new experiments and data, and now support publication of your manuscript.

There are only a few suggestions for minor corrections (made by referee #3), which you are kindly requested to address in a final version of your manuscript.

From the editorial side, there are also some changes and corrections that we need from you before we can proceed with acceptance of the manuscript:

- Please make sure that all deposited datasets mentioned in your Data Availability statement are now made publicly available and provide specific URLs for each one of them. The temporary link that was previously used for giving access to the referees can now be removed from this statement.
- Text coloring/highlighting and change tracking can now be removed from the main manuscript file.
- Please note that each co-corresponding author must link their ORCID ID to their profile in our manuscript handling system (two co-corresponding authors have not supplied their ORCID IDs yet).
- Please provide a list of up to 5 keywords after the Abstract of your revised manuscript.
- The author contributions statement should be removed from the manuscript file. Instead, we now use CRediT to specify the contributions of each author in the journal submission system. Please use the free text box to provide more detailed descriptions during submission. See also our guide to authors for more information:
<https://www.embopress.org/page/journal/14602075/authorguide#authorshipguidelines>.
- We noticed that callouts for Fig. 2D and Appendix Table S1-S2 are missing. Please make sure that all Figure panels and Appendix items are called out, in alphabetical order, in your manuscript.
- Please add the heading "Appendix" and the manuscript's title on the first page of your Appendix PDF file.
- Please also remove any text colors from the legends of your Appendix, as this file will be published online as supplied by you.
- We noticed that Source Data for Fig. 1A, 1C, 1E seem to be missing; please include all requested Source Data in your resubmission, along with the completed Source Data checklist that my colleagues have sent you.
- Please note that EMBO press papers are accompanied online by:
 - A) a short (2 sentences) summary of the findings and their significance,
 - B) 2-5 short bullet points highlighting the key results, and
 - C) a synopsis image in .jpg or .png format that is exactly 550 pixels wide and 300-600 pixels high (the height is variable within this range). You can either show a model or key data in the synopsis image. Please note that the text needs to be legible at the final size.Please upload this information along with your revised manuscript (the text for A and B should be provided in a separate Word file).
- Please note that the exact p values should be provided in the legends of Figures 1f-i; 5e.
- Please note that in Figure 5e there is a mismatch between the annotated p values in the figure legend and the annotated p values in the figure file that should be corrected.
- Please note that information related to "n" is missing in the legend of Figure 5d.
- Please note that the error bars are not defined in the legend of Figure 5d.
- The manuscript section order should be corrected: title page with complete author information, abstract, keywords, introduction, results, discussion, materials and methods, data availability section, acknowledgements, disclosure and competing interests statement, references, main figure legends, tables, expanded figure legends.

Please also note that as part of the EMBO publications' Transparent Editorial Process, The EMBO Journal publishes online a Peer Review File along with each accepted manuscript. This File will be published in conjunction with your paper and will include the referee reports, your point-by-point response and all pertinent correspondence relating to the manuscript. You can opt out of this by letting the editorial office know (contact@embojournal.org). If you do opt out, the Peer Review File link will point to the following statement: "No Peer Review File is available with this article, as the authors have chosen not to make the review process public in this case."

We look forward to seeing a final version of your manuscript as soon as possible. Please use this link to submit your revision: <https://emboj.msubmit.net/cgi-bin/main.plex>

Best regards,

Ioannis

Referee #1:

Authors did a good job and addressed all my concerns.

Referee #2:

The authors did an excellent job of addressing the reviewers' concerns. I have no additional comments prior to publication.

Referee #3:

The revised version of the report by Liu and colleagues properly answer all of my concerns. The authors performed an in-depth revision, adding new experimental and in silico data, and providing extra comparative analyses of the structures.

Super minor suggestions / Typos:

line 154 "Abundant of" might not be necessary at the beginning of the sentence

line 157 "state" should be plural

line 169 "is similar" maybe plural ?

line 196. Most of the time, when people use the prime numerotation, they put the amino acid first, eg L9'.

Figure 3D, maybe depict only the zone in y that corresponds to the TMD? E.g. 20-70, similar to what is done in panel B. Such a zoomed view would help the reader locate the hydrophobic gate easily. The results are interesting as it seems that for apo and betaine, the pore is close to be hydrated, with some relaxation to an hydrated form in the case of betaine-bound

All editorial and formatting issues were resolved by the authors.

Dear Qingfeng,

Congratulations on an excellent manuscript, I am very pleased to inform you that it has been accepted for publication in The EMBO Journal. Thank you for your comprehensive responses to the referee concerns and our editorial requests.

If you have any questions, please do not hesitate to contact the Editorial Office. Thank you for your contribution to The EMBO Journal. It has been a pleasure working with you!

Best wishes,

Ioannis
